# Liquid-in-liquid printing of 3D and mechanically tunable conductive hydrogels

Xinjian Xie [1], Zhonggang Xu[1], Xin Yu[2], Hong Jiang [2], Hongjiao Li [3] ✉ & Wenqian Feng [1,4] ✉

Conductive hydrogels require tunable mechanical properties, high conductivity and complicated 3D structures for advanced functionality in (bio) applications. Here, we report a straightforward strategy to construct 3D conductive hydrogels by programable printing of aqueous inks rich in poly(3,4-ethylenedioxythiophene):poly(styrene sulfonate) (PEDOT:PSS) inside of oil. In this liquid-in-liquid printing method, assemblies of PEDOT:PSS colloidal particles originating from the aqueous phase and polydimethylsiloxane surfactants from the other form an elastic film at the liquid-liquid interface, allowing trapping of the hydrogel precursor inks in the designed 3D nonequilibrium shapes for subsequent gelation and/or chemical cross-linking. Conductivities up to 301 S m$^{-1}$ are achieved for a low PEDOT:PSS content of 9 mg mL$^{-1}$ in two interpenetrating hydrogel networks. The effortless printability enables us to tune the hydrogels' components and mechanical properties, thus facilitating the use of these conductive hydrogels as electromicrofluidic devices and to customize near-field communication (NFC) implantable biochips in the future.

Electronics, such as brain-computer interfaces, are one of the most important materials for connecting electronic functionality and biological systems[1-4]. In particular, flexible bioelectronics have attracted extensive attention owing to their tremendous potential in disease treatment, such as Parkinson's or amyotrophic lateral sclerosis[5-8]. Relative to most conventional rigid electronics that are physically and mechanically dissimilar to biological tissues, conductive hydrogel electronics offer both favorable electrical properties and ideal interfaces with the tissues, eliminating immunological responses or the electrochemical instability arising from severe mechanical mismatch[9-11]. To mimic the mechanical properties of a wide range of biological tissues from ultrasoft to stiff (with elastic modulus values ranging from 0.1–500 kPa)[12], conductive hydrogels are required to form mechanically compliant interfaces without compromising their electronic performance. Additionally, the prevailing fabrication of conducting hydrogels has mostly relied on conventional manufacturing techniques for 2D patterns[13-15]. To broaden the applications of conductive hydrogels, a transformation of customized hydrogel (bio)

electronics from traditional 2D thin films to shape-conformable, integrated 3D structures is in process[16-20].

Among the investigated conductive polymers used to make electrically conducting hydrogels, poly(3,4-ethylenedioxythiophene):poly(styrene sulfonate) (PEDOT:PSS) has many advantages owing to its biocompatibility, water solubility, intrinsic electronic conductivity and commercial availability[21-24]. With the aid of 3D printing technology that has emerged in recent years, PEDOT:PSS-based hydrogels with 3D microscale structures have been fabricated[25]. Reported examples of 3D PEDOT:PSS hydrogels either rely on inkjet extrusion of conducting polymer inks layer by layer[16,26,27] or the employment of light-based 3D printing where lasers or light-emitting diodes (LEDs) initiate spatial photopolymerization in liquid[28-33]. Unfortunately, the inject extrusion method has a specific rheological requirement for the employed polymer inks, which brings difficulties in ink preparation and thus makes it difficult to selectively tune the mechanical properties of the formed gel. In the light-based 3D printing process, the light-induced curing of a prepolymer or monomer

[1]College of Polymer Science and Engineering, Sichuan University, 610065 Chengdu, China. [2]Department of Pancreatic Surgery, Department of Biotherapy, West China Hospital, Sichuan University, 610065 Chengdu, China. [3]College of Chemical Engineering, Sichuan University, 610065 Chengdu, China. [4]State Key Laboratory of Polymer Materials Engineering, Sichuan University, 610065 Chengdu, China. ✉e-mail: hongjiao.li@scu.edu.cn; feng.wenqian@scu.edu.cn

mixture first forms an inert hydrogel matrix. Then, the conductive components are introduced in the hydrogel system by blending PEDOT:PSS with hydrogel-forming precursors before curing[30,31] or by in situ polymerization of EDOT within the resulting hydrogel matrix[32,33]. Such disconnected aggregates of the conductive polymer network result in a high number of open circuits, making both strategies suffer from low conductivity (0.01–2.2 S m[-1]), even with a relatively high PEDOT:PSS content[34]. Additionally, constructing hydrogels with 3D overhanging structures does not accommodate soft objects in the absence of auxiliary support.

Liquid-in-liquid 3D printing is a promising approach that allows for the freeform fabrication of soft materials[35]. In this technique, the ink phase is extruded into a bath phase while a translation stage moves according to a 3D design, shaping the soft material into complex structures. However, suppressing Rayleigh-Plateau instabilities and minimizing the deformation of the extruded ink in the bath phase remain significant challenges in this process. Unlike embedded extrusion 3D printing, which requires specific support bath phases with shear-thinning or solid-fluid transition capabilities[36,37], another liquid-in-liquid printing technology utilizes self-assembly[38], jamming[39], and coacervation[40] to stabilize the filament extruded from the jet nozzle. This approach partly eliminates the need for rheological properties in both ink and support bath phases. Despite these advantages, reports on the use of liquid-in-liquid 3D printing for PEDOT:PSS conductive hydrogels are rare. While some cases of fibrillary gelation of PEDOT:PSS by injection into coagulation baths[40–42] have been reported, the printing method still requires strict matching of ink concentration, support bath components, and extrusion rate due to

the limited ability to overcome Rayleigh-Plateau instabilities. Although considerable efforts have been devoted to 3D printing conductive polymers, constructing 3D hydrogels with tunable mechanical properties, high conductivity, a broad selection of ink materials, and complicated structures remains challenging.

To address these limitations, we describe a liquid-in-liquid 3D printing process for the production of highly conductive PEDOT-based hydrogels with tunable stiffness and arbitrary structures (Fig. 1). Relying on the interfacial jamming of PEDOT:PSS–PDMS surfactants (PPSs) at the immiscible aqueous and oil-liquid interface, this process provides compelling routes to trap the aqueous inks into a nonequilibrium and 3D programmable shape and physically and/or chemically convert the liquid ink into the hydrogel state inside the defined profile. As such, the mechanical stability of the PEDOT:PSS–surfactant elastic film enables us to implement PEDOT:PSS physical gelation to form a loose gel interpenetrated by a secondary and robust polymer network. Conductivities up to 301 S m[-1] are achieved for a low PEDOT:PSS content of 9 mg mL[-1]. Importantly, the printable PEDOT:PSS inks are highly versatile, covering a broad range of PEDOT:PSS concentrations (in this study, the investigated concentration is 0.1–20 mg mL[-1]) and viscosity values (the corresponding viscosity of aqueous inks ranges from 6.5–23000 mPa·s, with inks of higher viscosity also being applicable). Therefore, the mechanical properties of the cured hydrogels can be orthogonally controlled by tuning the ink formulation and cross-linking conditions. Given the printed hydrogels' compatibility with biological systems, such easy printability and designability would allow for the fabrication of stretchable hydrogels without compromising conductivity, multimaterials integrated with

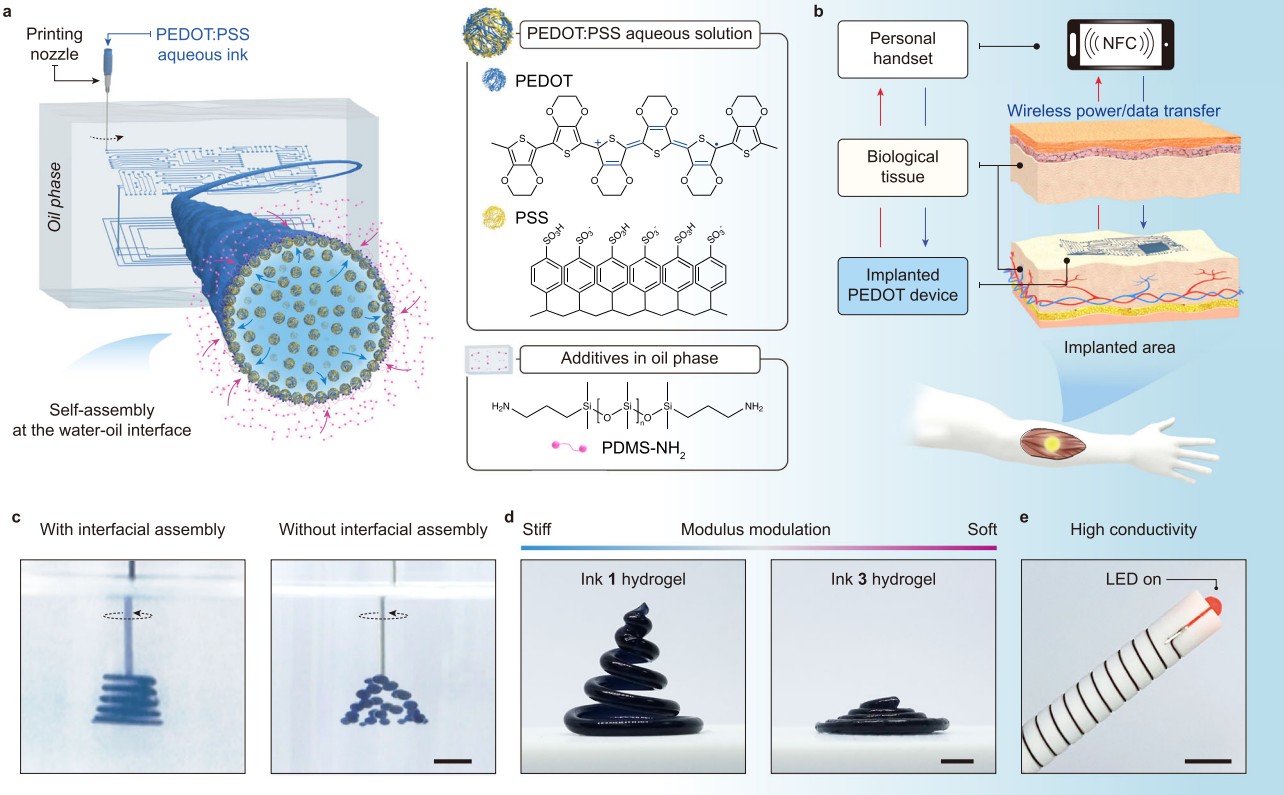

**Fig. 1 | Liquid-in-liquid 3D printing of hydrogels with tunable morphology, mechanical properties and conductivity. a** Schematic of the 3D printing of PEDOT:PSS-based aqueous threads in an oil. PEDOT:PSS–PDMS surfactants self-assemble at the liquid-liquid interface, forming an elastic wall that allows the liquid ink architecture to maintain integrity for the subsequent treatments. **b** Schematic of the soft PEDOT device for wireless sensing and simulation. **c** Photographic

images showing the liquid-in-liquid 3D print with and without interfacial PPSs assembly. The scale bar is 3 mm. **d** Photographic images showing the cured hydrogels with different stiffnesses. The scale bar is 3 mm. **e** The combination of high elasticity and conductivity of the hydrogel allows effective current transmission under arbitrary deformation. The scale bar is 1 cm.

3D hydrogels, electrochemical devices and, notably, conductive hydrogels for implantable chips.

## Results

### PEDOT:PSS-PDMS surfactants interfacial assembly

The typical conformation of PEDOT:PSS in water is expected to be colloidal particles that consist of a hydrophilic PSS shell surrounding a hydrophobic PEDOT core region[43]. Although PEDOT:PSS in the aqueous phase alone does not possess significant interfacial activity (Supplementary Fig. 1), the addition of a PDMS surfactant terminated with primary amines (PDMS-NH$_2$) to an immiscible oil solution leads to the self-assembly of PEDOT:PSS colloidal particles and PDMS-NH$_2$ surfactants at the water-oil interface, forming a jammed PEDOT:PSS-surfactants (PPSs) composite (Fig. 2a). The rapid interfacial assembly can be assumed to be a result of electrostatic interactions[44–46] between the exposed sulfonate groups (R-SO$_3^-$) in the PSS-rich shell and protonated amino groups (PDMS-NH$_3^+$) from the oil phase at the liquid-liquid interface. Since the two components can only exist in their respective phases, strong electrostatic adsorption causes the jammed assembly to become immobilized at the liquid-liquid interface, arresting further shape changes in the drop and kinetically trapping the drop into a shape even far removed from equilibrium.

To demonstrate that the expected PPSs assembly behavior occurs at the interface, we investigated the co-assembly kinetics of PEDOT:PSS dispersion and PDMS surfactant using pendant drop tensiometry. This technique provides the time evolution of the interfacial tension (IFT) by fitting the axisymmetric profile of the droplet to the Young-Laplace equation[35]. The pH dependence of the sulfonate-ammonium ion-pairing was firstly investigated by monitoring the IFT of pendant droplets of a diluted commercial PEDOT:PSS dispersion (0.50 mg mL$^{-1}$) in a toluene phase consisting of bis(3-aminopropyl)-terminated PDMS-NH$_2$ ($M_n$ 4000 g mol$^{-1}$, 10 vol%) while varying the pH of the aqueous phase. We found that the rate of interfacial self-assembly of PPSs increases with decreasing pH (Fig. 2b and Supplementary Fig. 2). At higher pH values (>11), PDMS-NH$_2$ loses its ability to assemble at the interface with PEDOT:PSS particles due to deprotonation. Within the range of pH values from 3.01 to 10.87, the IFT of PEDOT:PSS droplets decreased below 18 mN m$^{-1}$, and well-defined wrinkles were observed at the interfacial area when extracting the aqueous droplets, revealing the irreversible formation of elastic films at the liquid-liquid interface. With further decreasing the pH until the pKa of PSS (PSS is assumed to have an average pKa of approximately 1.5[47]), PEDOT:PSS tends to exhibit a higher negative charge (Supplementary Fig. 3 and Supplementary Table 1). This results in stronger electrostatic interactions between the PEDOT:PSS particles and surfactants in such pH conditions. For pH values less than 3 or for higher PEDOT:PSS concentrations, the initial adsorption of PEDOT:PSS particles to the interface at pH 2.55 is therefore too rapid to form regular droplet shapes, preventing the measurement of IFT using the shape-fitting method. More specifically, adjusting the pH to 2.07 can lead to a continuous jet without breakup when the flow rate is set to 0.07 mL min$^{-1}$ (Supplementary Fig. 4). When the pH was further adjusted to 1.21, the PEDOT:PSS droplet became opaque, and parts of the assembly containing PEDOT:PSS were removed from the aqueous phase into the oil phase. This phenomenon may have been caused by the formation of micelles (water in oil (W/O)) at the liquid-liquid

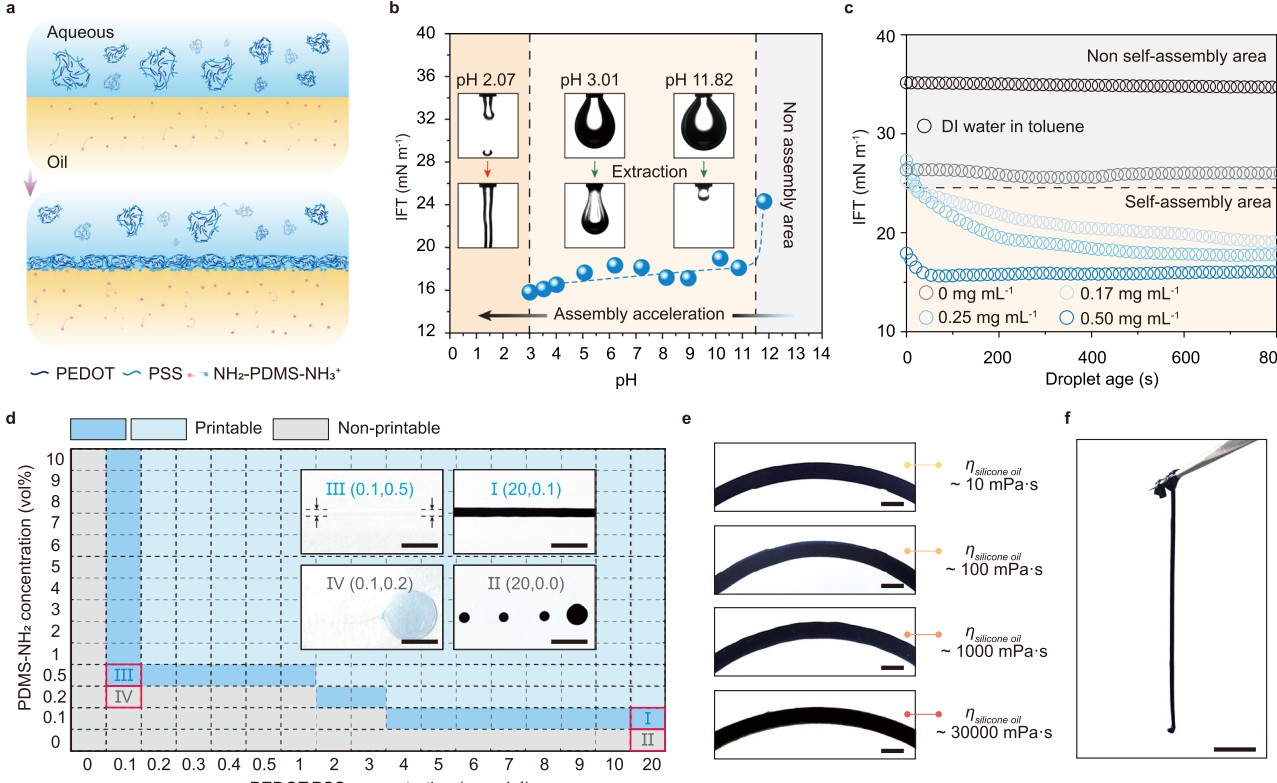

**Fig. 2 | PPSs interfacial assemblies for printing of a PEDOT:PSS aqueous dispersion in oil. a** Schematic of the interfacial assembly of PPSs and the elastic films they form. **b** Equilibrium interfacial tension (IFT) versus pH showing the pH-dependent assembly of PPSs at the water/toluene interface, as well as the droplet morphology and its buckling behavior when the PEDOT:PSS dispersion is withdrawn. **c** Temporal evolution of IFT of PEDOT:PSS aqueous dispersions (0, 0.17, 0.25 and 0.5 mg mL$^{-1}$) introduced to solutions of PDMS-NH$_2$ in toluene (10 vol%), illustrating control over the rate of PPSs assembly at the liquid interface. **d** Suitable concentration profiles of PEDOT:PSS aqueous dispersion and PDMS-NH$_2$ surfactant for liquid-in-liquid printing, showing the wide range of printing conditions. The scale bar is 500 μm. **e** Printing the PEDOT:PSS aqueous dispersion (10 mg mL$^{-1}$) in silicone oil with various viscosities. The scale bar is 2 mm. **f** The mechanically robust PPSs assembly allows the liquid thread of pure PEDOT:PSS dispersion to hang in air. The scale bar is 1 cm.

interface, which could be attributed to the very low interfacial tension and osmotic pressure at low pH levels[48,49]. These results indicate that there is a broad pH range from 2.07 to 10.87 that is compatible with this PEDOT:PSS-surfactant self-assembly system.

The concentrations of PEDOT:PSS in the aqueous phase influence the rate of diffusion to the interface and the areal density of polymers at the interface, which ultimately dictate the IFT and the mechanical strength of the assembled film at the steady state. As shown in Fig. 2c, increasing PEDOT:PSS concentration from 0.17 mg mL$^{-1}$ to 0.5 mg mL$^{-1}$ decreased the equilibrium time of IFT from 800 s to 40 s. When increasing the PEDOT:PSS concentration to more than 1 mg mL$^{-1}$, the system reached its steady state on a time scale of milliseconds, as evidenced by the behavior of instantaneous particle-surfactant self-assembly and maintenance of the droplet's original shape for at least 12 h (Supplementary Fig. 5). The formation of the PPSs interfacial assembly was ultimately confirmed through the fabrication of a planar layer of PPSs using a mold, as shown in Supplementary Fig. 6. SEM images revealed the densely packed PEDOT:PSS nanoparticles at the assembly layer, thus confirming our proposed interface assembly hypothesis.

Since the jammed assembly has the ability to counteract the compressive force in the system arising from Rayleigh-Plateau instabilities in a timely manner[39], it is convenient to use PPSs films to stabilize the aqueous architectures that are generated in the oil phase via direct-write methods (namely, liquid-in-liquid 3D printing). By moving the print head of a commercial extrusion-based 3D printer equipped with a syringe pump, the aqueous PEDOT:PSS dispersion was injected into the surrounding oil phase containing PDMS-NH$_2$ surfactant along the programmed trajectories, drawing the liquid ink into linear or curved threads (in this study, the term 'threads' refers to liquid inks with a thread-like shape and lacking any crosslinked structure). The printing conditions under which stable structures can form in silicone oil ($\eta_e$ ~ 30000 mPa·s) are summarized in the printability phase diagram (Fig. 2d). When the PEDOT:PSS and PDMS surfactants are beyond their critical concentrations, mechanically stable interfacial assemblies are obtained to prevent the threads from breaking up into droplets. Due to the quick assembly, the oil phase, even with a low viscosity ($\eta_e$ ~ 10 mPa·s), is also capable of quickly causing the PPSs to form threads from the liquid (Fig. 2e and Supplementary Fig. 7). Thus, the oil phase has minimal requirements, with only water immiscibility and good solubility to PDMS surfactants being necessary. The viscosity and rheological properties of the oil phase have little impact on the PPSs interfacial assembly process. For long-term printing, the density of the oil phase can be adjusted to match that of the aqueous phase to prevent the printed aqueous structures from settling under gravity. Such interfacial assembly is surprisingly solid in that the printed aqueous threads manage to retain their structures for at least 12 h (Supplementary Fig. 5) and can even be pulled out of the oil without rupturing (Fig. 2f and Supplementary Fig. 8). All these features allow extensive manipulation of the aqueous threads within the physical framework and within the experimental timeframe.

## Printing PEDOT-based hydrogels with a chemically crosslinked network

In a first demonstration of 3D printing a conductive hydrogel, we mixed the hydrogel precursor, poly(ethylene glycol) diacrylate ($M_n$ 1000 g mol$^{-1}$, PEG 1000), and blue-light photoinitiator, lithium phenyl-2,4,6-trimethylbenzoylphosphinate, into the pristine PEDOT:PSS dispersion to prepare polymer ink 1 (Table 1). After drawing the liquid ink into threads in the surrounding oil phase, subsequent light irradiation ($\lambda = 395$ nm) initiated the radical polymerization of PEG diacrylate to generate the chemically crosslinked network of the PEDOT@PEG hydrogel (Fig. 3a, b). The scanning electron microscopy (SEM) image clearly shows that this PEDOT@PEG hydrogel has a 5 μm-thick, wrinkled skin surrounding a cured PEG core (Fig. 3c), demonstrating that

the UV curing occurred throughout the whole thickness. The formation of the skin-core structure can be attributed to the differences in PEDOT:PSS particle density and PEG monomer concentration, which result from the PPSs interfacial assembly (Supplementary Fig. 9). This provides additional evidence for the liquid-liquid interfacial assembly that contributes to the hydrogel formation in desired shapes.

As the UV-induced chemical crosslinking of PEG-diacrylate plays an essential role in the mechanical reinforcement of ink 1 (Fig. 3d), the mechanical strength of the printed hydrogels can be easily tuned by varying the UV curing time, monomer concentration (Supplementary Fig. 10) and the selection of hydrogel-forming precursors. To test the compatibility of this system, five cylindrical PEDOT:PSS hydrogels (diameter: 1.5 mm) were fabricated by printing the inks 1–5 listed in Table 1. In all cases, the interfacial assemblies reached their steady state rapidly for subsequent UV curing. The hydrogels printed with different ink formulations and curing conditions exhibit distinct transitions in mechanical characteristics, transitioning from stiff (Young's modulus: 3875 kPa) to soft (Young's modulus: 41.2 kPa) (Fig. 3e). As such, the range of hydrogels that can be generated by this printing technique is rather broad as long as the hydrogel-forming precursors can coexist with PEDOT:PSS microgel particles in dispersion. The ability to control mechanical properties using the cross-linked hydrogel network enables us to take advantage of the stiffness or elasticity of certain hydrogel formulations to impart the designed mechanical strength onto the conductive composite. As a preliminary demonstration, the PEDOT@PAAM/PEG hydrogel (ink 4) is able to keep a light-emitting diode (LED) lit even after being stretched to 800% strain (Fig. 3f).

Next, we quantified the electrical conductivity of these printed hydrogels by measuring the resistance of the gels using a two-point probe method (Table 1). The conductivity of the hydrogels tends to increase as a function of PEDOT:PSS concentration in the hydrogel precursor solution (Supplementary Fig. 11). The increase in PEDOT:PSS concentration over 20 mg mL$^{-1}$ in PEDOT@PEG gel, however, results in a sharp rise in the apparent viscosity of the polymer inks[16]. The tendency for PEDOT:PSS particles to agglomerate in these viscous inks further leads to an increase in printing head-clogging risk, particularly in the cases where small nozzles are used. Thus, to improve the electrical properties of our hydrogels that were printed with low PEDOT:PSS concentrations, we conducted a post-treatment process (i.e., soaking the hydrogels in ethylene glycol and dry annealing at 130 °C) once the printed inks were solidified. The conductivity of hydrogels made from ink 5 (9 mg mL$^{-1}$ PEDOT:PSS content) was enhanced by 3 orders of magnitude after post-treatment (ink 5*, 61.8 S m$^{-1}$ in the dry state and over 1.9 S m$^{-1}$ in the hydrogel state, Supplementary Fig. 12). Besides the known mechanisms of phase separation of PEDOT and PSS during the soaking process to form a denser PEDOT network[50,51] and the recrystallization of PEDOT-rich domains during dry annealing to generate higher conductive pathways[52,53], the water-loss-induced collapse of the conducting PEDOT@PEG network into aggregates during the dry annealing process is another significant factor that contributes to the enhanced conductivity of the printed gels, as revealed by the comparison of different drying treatments on printed hydrogels (Supplementary Fig. 13). The denser PEDOT network of the gels in the dry state also highly improved the modulus of the gels to dissipate more stress from mechanical deformation (Fig. 3e).

Interestingly, we found that the resistance of the post-treated gels made from ink 5* gradually decreased during the stretching-releasing process (Fig. 3g). Reversible stretching of the gel between 0 and 50% strain is likely to encourage a better orientation and more π–π stacking interactions between the PEDOT networks, which contribute to lowering the gel's resistance by 12% after 30 cycles. Supporting this finding, grazing-incidence wide-angle X-ray scattering (GIWAXS) measurements show that the PEDOT:PSS crystallites in the gel changed from a disordered arrangement to a preferred face-on molecular

**Table 1 | Different formulations and processing of PEDOT hydrogels and their corresponding electronic and mechanical properties**

| Formulation number | PEDOT:PSS concentration (mg mL⁻¹) | Additive monomer | Monomer concentration (mg mL⁻¹) | Ionic liquid concentration (wt%) | UV$_{\lambda=395\,nm}$ irradiation time (min) | Post-treatment (Y/N) | Modulus (kPa) | Strain at break (%) | Conductivity (S m⁻¹) |
|---|---|---|---|---|---|---|---|---|---|
| Ink **1** | 5 | PEG 1000 | 385 | – | 5 | N | 3875.0 | 76 | $1.0 \times 10^{-2}$ |
| Ink **2** | 5 | PEG 6000 | 455 | – | 5 | N | 195.2 | 186 | $9.2 \times 10^{-3}$ |
| Ink **3** | 5 | PEG 20000 | 313 | – | 5 | N | 99.0 | 287 | $8.5 \times 10^{-2}$ |
| Ink **4** | 5 | AAM/ PEG 20000 | 263/263 | – | 5 | N | 161.5 | 800 | $1.9 \times 10^{-2}$ |
| Ink **5** | 9 | PEG 1000 | 100 | – | 2 | N | 41.2 | 136 | $5.6 \times 10^{-2}$ |
| Ink **5**$^{*}$ | 9 | PEG 1000 | 100 | – | 2 | Y | 685.0 | 127 | 61.8 |
| Ink **6** | 9 | PEG 1000 | 100 | 2 | 2 | N | 33.2 | 96 | $6.8 \times 10^{-1}$ |
| Ink **6**$^{*}$ | 9 | PEG 1000 | 100 | 2 | 2 | Y | 54.4 | 130 | 301.9 |
| Ink **7** | 9 | PEG 20000 | 100 | 2 | 2 | N | 7.3 | 263 | $2.0 \times 10^{-1}$ |
| Ink **7**$^{*}$ | 9 | PEG 20000 | 100 | 2 | 2 | Y | 26.8 | 275 | 163.4 |

All hydrogel samples were printed using a $\Phi$1.6 mm nozzle with 0.5 m min⁻¹ print head speed and 1.0 mL min⁻¹ ink flow rate.

All hydrogel samples were printed in oil phase (10 vol% PDMS-NH₂ in silicone oil, $\eta_e \sim 30000$ mPa·s).

orientation[54] relative to the stretching direction during stretching. Furthermore, a red shift of the $C_\alpha = C_\beta$ vibration peak indicates that the cyclic stretching process leads to a more planar backbone of PEDOT, which contributes to more efficient charge delocalization and a higher packing order[55,56] (Supplementary Fig. 14).

We further investigated the electrical stability of these printed gels. Compared with the chemically cured PEDOT@PEG hydrogel ink 5, cyclic voltammetry (CV) measurements show that the post-treated ink 5* hydrogel has satisfactory electrochemical stability against charging and discharging cycles in a wet environment (10× phosphate-buffered saline) with no obvious shifts in the oxidization and reduction peaks after 1000 CV cycles (Fig. 3h, i). Despite the absence of a covalent bond or strong physical interaction between the PEDOT chain and PEG network, the PEDOT gradually migrates away from the hydrogel network and into the surrounding aqueous environment after a period of more than two months (Supplementary Fig. 15). These elastic and conductive gels with a single chemically crosslinked network are more suitable for practical applications in air.

**Highly conductive PEDOT-based hydrogels with two interpenetrating networks**

Macroscopically connecting PEDOT:PSS particles as a conductive network to increase the number of charge conduction pathways is regarded as another effective strategy to improve the conductivity of PEDOT:PSS hydrogels[57,58]. To generate a sufficient number of physical crosslinks between particles, we first mixed the ionic liquid 1-carboxymethyl-3-methylimidazolium bis(trifluoro-methylsulfonyl) imide (HOOCMIMNTF2) (2 wt%) in printing inks 6 and 7 and extruded the ink into threads suspended in the oil phase with PDMS-NH₂. Since ionic liquids are effective in screening the electrostatic repulsions between particles and thus triggering PEDOT:PSS to form physical crosslinks (Supplementary Fig. 16) through π–π stacking of PEDOT[59], these liquid inks trapped inside the PEDOT:PSS-surfactant assembly were gradually gelled into the designed 3D shapes (Fig. 4a, b). Rheological measurements (Fig. 4c) show that the physical crosslinking was triggered by the time the solution was mixed with HOOCMIMNTF2. Even though the storage modulus (G′) of the ink exceeded the loss modulus (G″) throughout the entire process, at this ionic liquid concentration, the ink mixture could maintain its flowability in the first 8 min (Supplementary Fig. 17). Since the gelation rate could be tuned by the concentration of the ionic liquid additive, for those processes that require a longer operation time, a lower concentration of HOOCMIMNTF2 could be selected to gel PEDOT:PSS at a slower rate.

Although the PEDOT:PSS networks (primary network) behave similarly to solids after 10 min of gelation, they are still highly brittle if manipulated further. An acrylate network (secondary network) was then consequently introduced into the PEDOT:PSS network by UV curing for 2 min (Fig. 4d), resulting in mechanical reinforcement of the resultant polymer matrix (Fig. 4e). The printed PEDOT@PEG hydrogels with two interpenetrating networks could be stretched to 96% elongation (PEG 1000 (ink 6)) and even up to 263% elongation (PEG 20000 (ink 7)) prior to fracture.

Due to the formation of PEDOT conductive networks with enhanced π–π stacking interactions for electron transfer, both the conductivity (Table 1) and electrical stability (Supplementary Fig. 15) of the printed hydrogels are largely improved. These hydrogels with post-treatments achieve an electrical conductivity as high as 301 S m⁻¹ (ink 6*), 4-fold higher than the value of another printed gel without ionic liquid treatment (ink 5*). Electrical behavior under mechanical deformation shows that the conductivities of the printed hydrogels remained nearly unchanged from their starting values despite large variations in tensile strains (50% strain for ink 6* and 200% strain for ink 7*) (Fig. 4f). In particular, there was no apparent change in the resistance of both ink 6* and ink 7* gels even after being stretched to 25% strain. The robust conductivity can be attributed to the rearrangement of PEDOT domains that stay connected with others during the stress deformation, rather than disrupting or insulating the conductive regimes[34]. Owing to the strain-independent conductivity of the gels, an LED can be kept lit after stretching ink 7* to 200% elongation. These results indicate that elasticity has been successfully incorporated into our conductive polymeric materials without sacrificing their electrical properties. The ability to orthogonally control the mechanical and electrical properties therefore enables us to impart the required stiffness onto the highly conductive composite.

**Conductive hydrogel with 3D structures**

Conducting polymer solutions, such as the pristine PEDOT:PSS aqueous dispersion (CLEVIOS™ PH 1000) used in this study, are typically not suitable for direct use in extrusion-based 3D printing or inkjet printing due to their fluidity. A significant advantage of using PEDOT:PSS-surfactant films to stabilize the liquid hydrogel precursor is that this printing technique refrains from restricting the viscosity and yield stress of the printing ink. As shown above (Fig. 2d), PEDOT:PSS inks with concentrations from 0.1 mg mL⁻¹ to 20 mg mL⁻¹ are applicable for printing free-form hydrogels with complex architectures. The sizes of the ink threads can be readily modulated by adjusting the print

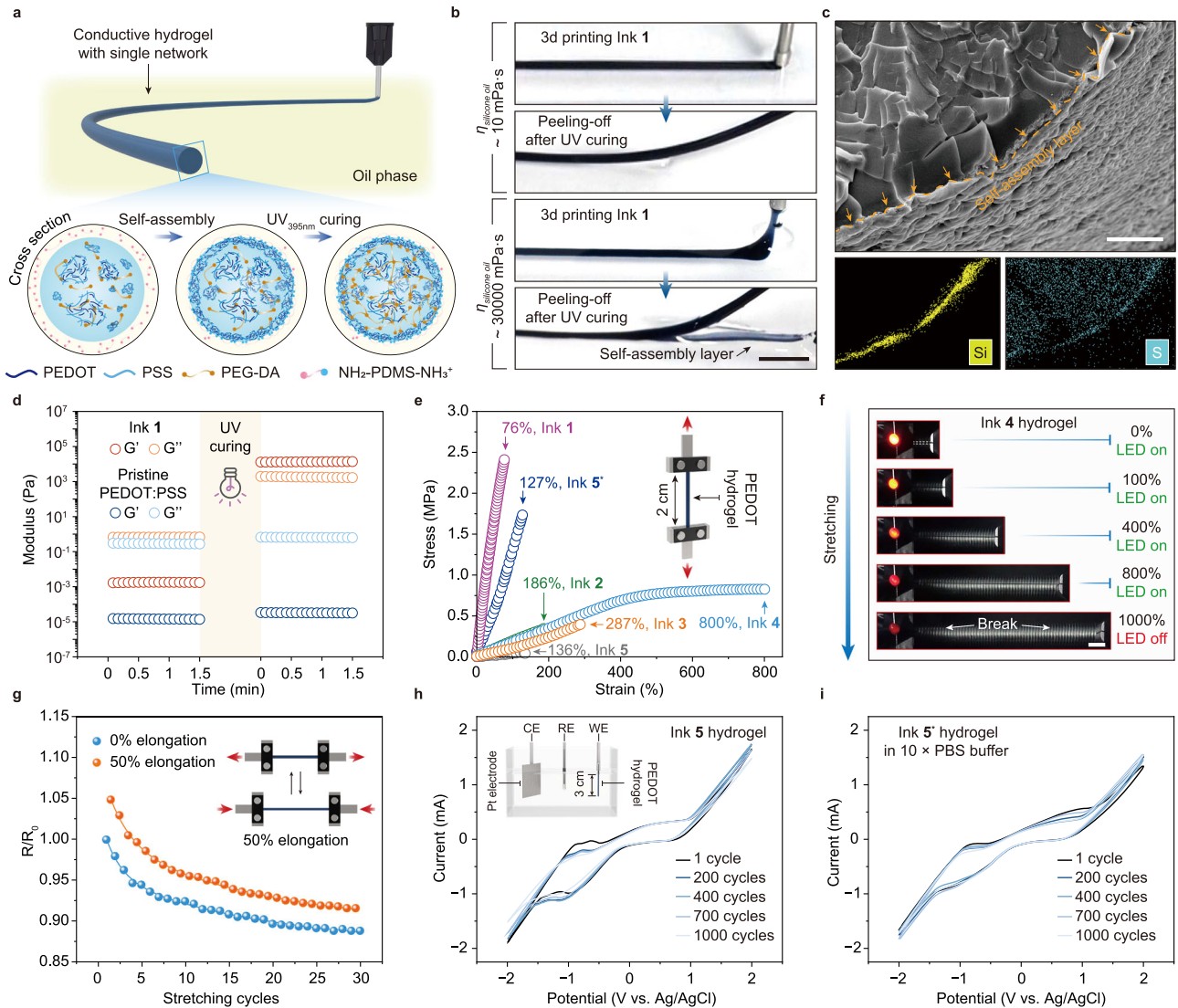

**Fig. 3 | Printed PEDOT hydrogels with a chemically crosslinked network.**
**a** Schematic of 3D printing of PPSs-stabilized hydrogel precursor threads in silicone oil with PDMS-NH₂ surfactants, and the PEDOT-based hydrogel formed after UV-curing the acrylate monomers. **b** Linear thread of printed ink 1 in silicone oils before UV curing into hydrogel. The PPSs assembly shell peels off in the viscous silicone oil. The scale bar is 5 mm. **c** Cross-sectional SEM image/EDS mapping of freeze-dried PEDOT@PEG (ink 1) showing that the final gel has a skin-core structure. Similar results were observed in two independent samples. The scale bar is 100 μm. **d** Rheology measurement showing the UV-induced chemical crosslinking of the

printing inks. **e** Tensile elongation curves of the gels with different ink formulations, showing that all formulations can be successfully printed despite large differences in the elastic moduli of the cured gels. **f** The high conductivity and high elasticity of the hydrogel (ink 4) keeping an LED lit even after being stretched to 800% strain. The scale bar is 1 cm. **g** Resistance change, expressed as a ratio between resistance (R) and initial resistance (R₀), across a dry-annealed gel (ink 5*) while it is strained reversibly between 0 and 50% for 30 cycles. **h** and **i** CV curves for the hydrogel without (ink 5, **h**) and with (ink 5*, reswollen state, **i**) post-treatment in PBS.

head speeds and flow rates. For instance, being equipped with a 14-gauge needle (internal diameter = 1.6 mm), the average diameter of the printed hydrogels varies from 2027 ± 64 μm to 94 ± 12 μm (Fig. 5a and Supplementary Fig. 18). The excellent uniformity and high yield of the PEDOT hydrogels (Fig. 5b) allow us to customize the conductive filaments facilely and accurately.

With the PEDOT:PSS-surfactant assembly at the liquid-liquid interface, the squeezed liquid threads can be physically isolated and prevented from sticking to or fusing with each other during the whole process. Thereby, we successfully fabricated a 1.1 m-long conductive gel in a 1.8 cm-diameter circle of printing area (Fig. 5c, d, and Supplementary Movie 1). As the viscous oil phase ($\eta_e$ ~ 30000 mPa•s) could restrain the ink threads and counteract the effect of gravity, overhung structures that cannot be achieved in extrusion-based 3D printing were finely realized in our print (Fig. 5e, Supplementary Fig. 19 and Supplementary Movie 2). The compatibility with various inks enables this

printing technique to integrate multiple materials into 3D hydrogels by programming the ink feed. A helix hydrogel consisting of two heterogeneous components was fabricated by sequential injection of distinct inks as a demonstration (Supplementary Fig. 20).

Apart from two of the existing strategies for 3D tubular structured hydrogels, i.e., the template method[60] and the digital light processing printing technique[61], our approach provides another solution for tubular hydrogel fabrication when the print head is replaced by a coaxial double needle. Herein, while injecting the aqueous ink into a silicone oil bath ($\eta_e$ ~ 30000 mPa s, oil phase 1) through the outer ring of the print head, another silicone oil solution of PDMS-NH₂ ($\eta_e$ ~ 1000 mPa•s, oil phase 2) was squeezed out from the interior cavity of such a needle at the same time (Fig. 5f). In this way, two concentric PPSs assembly layers were simultaneously formed at both the internal and external interfaces of the aqueous phase and thus shaped the ink into a tubular structure. The resulting hydrogel after chemical curing

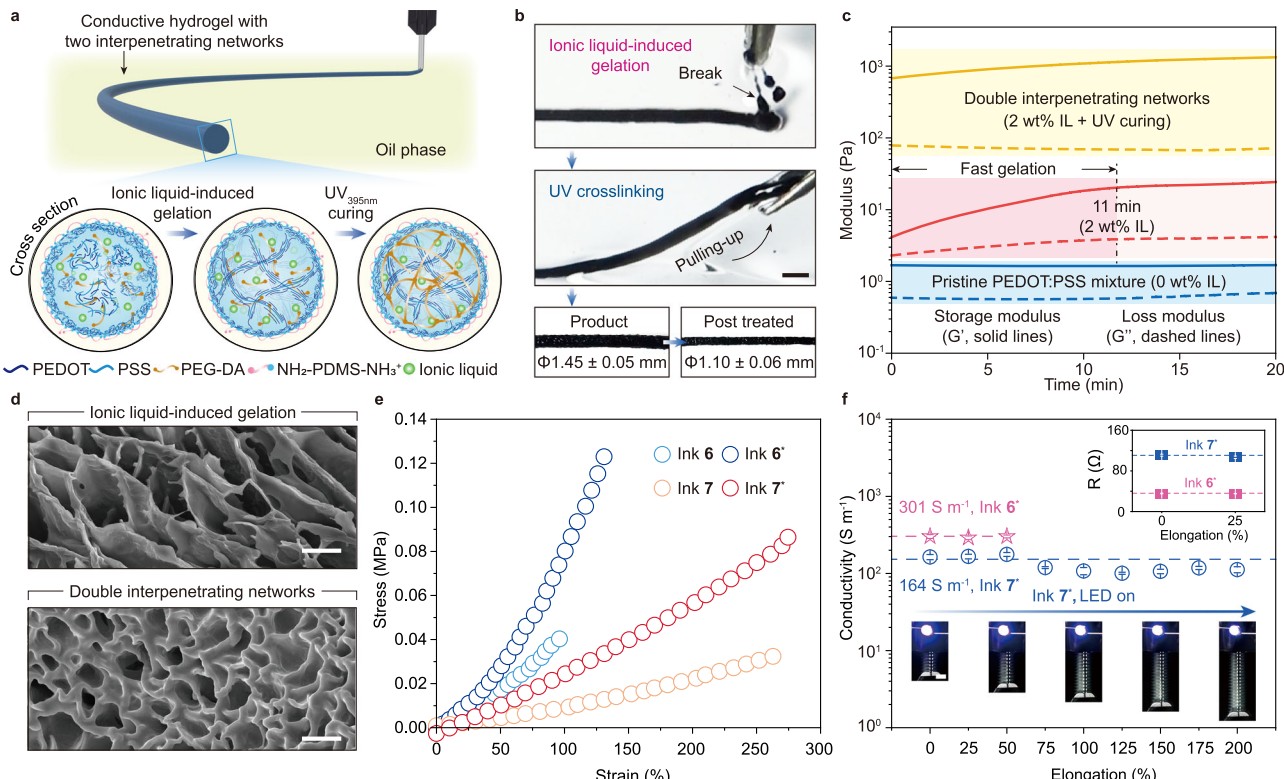

**Fig. 4 | Printed PEDOT hydrogels with two interpenetrating networks. a** Process for fabricating PEDOT interpenetrating hydrogel networks. First, PPSs-stabilized hydrogel precursor threads are 3D printed in silicone oil with PDMS-NH₂ surfactants. Second, PEDOT:PSS hydrogels are formed in the aqueous threads by ionic liquid-induced aggregation of the remaining PEDOT:PSS colloidal particles into a macroscopically connected network. Third, the polyacrylate network is chemically crosslinked inside the PEDOT:PSS hydrogels by polymerizing the acrylate monomers under UV light. **b** Fabrication of ink 6* hydrogel thread. The printed ink thread forms a brittle PEDOT:PSS gel after 10 min and is mechanically strengthened by subsequent chemical cross-linking. The scale bar is 3 mm. **c** Rheology measurement showing the ionic liquid-induced gelation and UV-induced chemical crosslinking of printing ink 6. **d** Cross-sectional SEM image of freeze-dried ink 6 hydrogels showing a single PEDOT:PSS network and two interpenetrating networks corresponding to each step. The similar results were observed across two independent samples within each group. The scale bar is 10 μm. **e** Tensile elongation curves of the ink 6 and ink 7 hydrogels before and after post-treatment. **f** Electrical conductivity of the ink 6* and ink 7* gels at different strains showing the strain-independent conductivity. The resistances of both gels remain constant near their initial value at 25% tensile strain. Due to the high and strain-independent conductivity, the hydrogel is able to keep an LED lit even after being stretched to 200% strain. The scale bar is 5 mm. Values in **f** represent the mean, and the error bars represent the SD of the three measured values of each sample (*n* = 3). The resistances and diameters of three replicate samples were measured in real-time while undergoing identical stretching processes in **f**.

presents a clearly concentric circle cross-section without any collapse, and both the aqueous and organic liquids could be smoothly pumped through the channel (Fig. 5g, h). These results reveal the ability to facilely control the 3D morphology and composite properties of the printed hydrogels.

## Hydrogel-based electrochemical microfluidic devices

To verify the capability of the printed tubular PEDOT:PSS hydrogel in creating an electric field of coaxial cylinders, we investigated the use of the printed 3D tube as an electro-microfluidic device[62] for conducting electrochemical reactions, such as copper electroplating[63]. As schematically shown in the working mechanism of the reaction in Fig. 6a, the hydrogel shell was designed to be connected with the positive electrode of a direct current (DC) source as an anode. A Pt wire, the object to be plated, was installed in the center of the hydrogel tube. To prevent the Pt wire from touching the hydrogel shell and produce a short circuit, the Pt wire was pinned through two electrically insulating O-ring gaskets that were placed at the inlet and outlet of the hydrogel channel. The gaskets were sealed tightly with the hydrogel shell to prevent leakage of the electrolyte during flow (Fig. 6b). Once the Pt wire was connected to the DC source to complete the circuit, we pumped a solution of 1 M CuSO₄ through the channel to trigger the electrolytic reaction. The real decomposition voltage in this fluidic

electrolytic cell was determined to be 1.36 V by measuring the V-I characteristic curve (Fig. 6c). While pumping the solution under given voltages, the reduction of Cu(II) ions in electrolyte flow resulted in the deposition of the desired elemental Cu(0) on the Pt cathode surface, as demonstrated by the cathode thickness variation and the SEM imaging associated with EDX measurements (Fig. 6d, e and Supplementary Fig. 21). The byproduct O₂ generated from water in the hydrogel shell was emitted out of the channel along with the fluid stream. The uniform copper coating on the surface of the platinum wire confirms the presence of an electric field created by coaxial cylinders.

Since the ink formulation plays an important role in tuning the conductivity of the printed gels, the rate of the electrolysis process is also affected. The electrolysis in a pure PEG hydrogel tube that was fabricated via the template method was quite slow (Fig. 6f), despite the diffusion of CuSO₄ electrolyte inside the hydrogel improving the conductivity of the tubular shell. In contrast, the printed ink 6* hydrogels could finish the electrolysis reaction within only four minutes, obtaining a 162 ± 41 μm thick copper layer at the Pt cathode. Although the conductivity of PEDOT:PSS tubular gels is inferior to that of metal tubes, their ease of printing and adjustable mechanical properties make them a promising candidate for producing coaxial cylindrical conductors, including customized coaxial cables for implantable patch antennas[64], in a more efficient and tailored way.

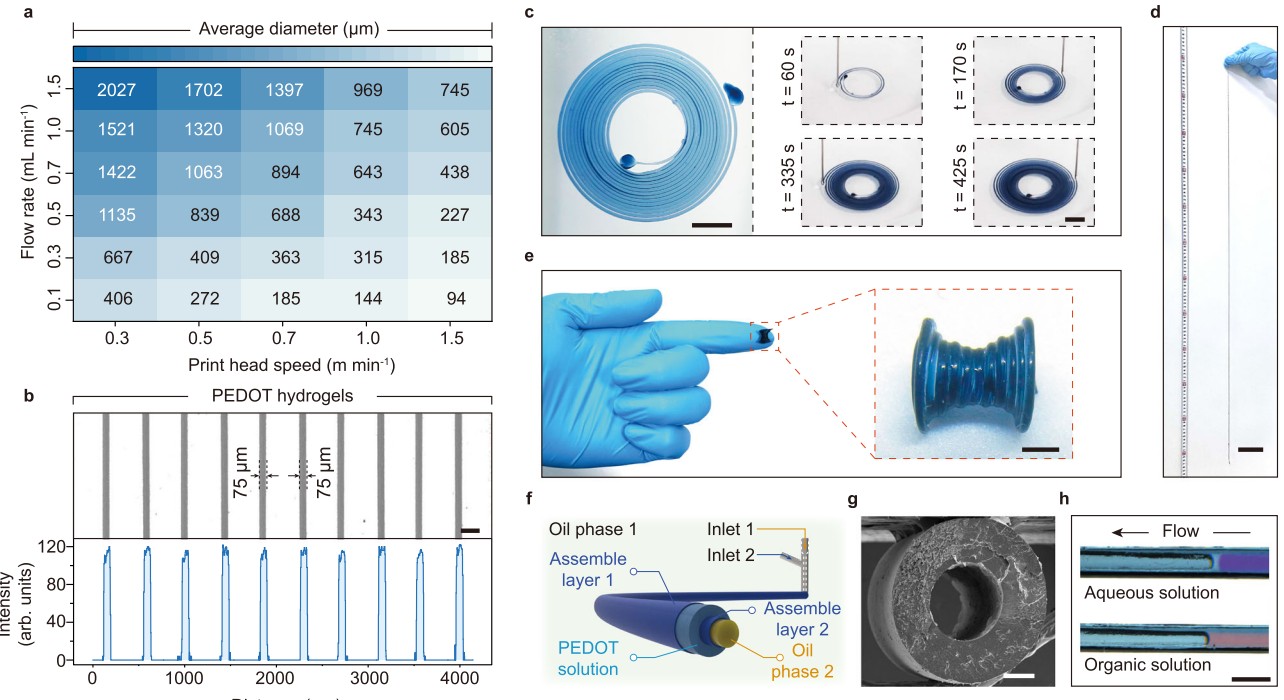

**Fig. 5 | Printing Conductive hydrogels with 3D structures. a** Diameter control of the printed hydrogels. The aqueous printing ink (ink 1) was injected at varying flow rates and print head speeds into the oil phase. **b** Top: optical image of PEDOT hydrogels 75 μm in diameter obtained by direct liquid-in-liquid printing. Bottom: intensity profile showing the high uniformity of the obtained hydrogels. **c** An aqueous ink spiral (ink 1), 1.1 m long with a thread thickness of 300 μm, in silicone oil. These liquid threads can be physically isolated with the PPSs to prevent them from sticking to or fusing with each other. Scale bar is 4 mm. **d** The fabricated ink 5* gel filament. The scale bar is 7 cm. **e** A printed PEDOT hydrogel (ink 1) in a 3D hourglass shape. The scale bar is 3 mm. **f** Schematic of 3D printing of conductive and tubular structured hydrogels enabled by double PPSs interfacial assembly layers. **g** Cross-sectional SEM image of freeze-dried hydrogels with tubular frames (ink 1). The scale bar is 300 μm. **h** Flow of an aqueous solution of methylene blue and a hexane solution of oil red through the tubular PEDOT hydrogel. The scale bar is 2 mm.

## PEDOT-based NFC chip

Near-field communication (NFC) is a short-range and wireless technology that allows simultaneous power and data transmission between devices, offering exciting opportunities for the design of miniature and battery-free sensing systems in health care[65]. Enabled by the effortless 3D printability as well as the combination of elasticity and electrical properties, 3D printing of the conducting polymer ink can offer a promising route for the facile fabrication of conducting polymer-based devices. As a proof-of-concept of the integration of these conductive gels into applicable devices, we fabricated a simple PEDOT-based NFC antenna at room temperature. As shown in Fig. 7a, we used the 3D printing method detailed in Fig. 5c to produce the filament-shaped PEDOT gels (ink 5*, approximately 1 meter long), which were then strung onto a chip holder that possessed a continuous coil groove to prevent contact short circuits between the conductive PEDOT gels (Fig. 7b).

The PEDOT-based NFC antennas show stable accessibility for various NFC-enabled equipment. The wirelessly harvested energy from a smartphone via the PEDOT NFC antenna was able to make the connected LED blink at a cycle of 2 Hz (Fig. 7c and Supplementary Movie 3). The amplitude of the radio frequency signal (13.56 MHz) from the smartphone can also be read and collected through the biological tissue (here, a 3-mm thick pork rind was used), showing that the wireless transfer of power and data through skin is possible. With the increase in the signal transmission distance, the current density in the circuit shows a declining trend (Fig. 7d).

To validate the biocompatibility of our printed PEDOT:PSS-based hydrogels for potential use as implantable materials, we performed experiments to assess their cytocompatibility in vitro and histological analysis in vivo. We found that after 72 h of culture, the tested hydrogel samples (ink 5, ink 5*, ink 6, and ink 6*) demonstrated high cell proliferation with a viability percentage of approximately 95% (Supplementary Fig. 22). Despite the changes in interfacial density of PEDOT:PSS particles and concentration of PEG monomers resulting from the PPSs interfacial assembly process, the printed PEDOT@PEG hydrogels exhibited anti-cell adhesion properties (Supplementary Fig. 23), which are critical for preventing implant materials from failing due to unpredictable protein adsorption and cell interaction. In addition, we observed high biocompatibility of the printed PEDOT@PEG hydrogels (ink 6* with the most components) in immunocompetent mice at a time point of 7 days (Supplementary Fig. 24), with unsacrificed mice remaining alive and healthy without any abnormalities for two months after subcutaneous implantation. Although we failed to implant this unencapsulated PEDOT NFC tag into biological tissue because of the inevitable short circuit in a wet physiological environment, the energy transmission capability suggests the great potential of these PEDOT-based gels to be applied in next-generation, battery-free, wireless sensing and electrical impulse delivery devices, such as artificial cardiac pacemakers[66] or devices for cranial electrotherapy stimulation[67]. Experiments for creating elastic and conductive gels surrounded with insulative shells for stretched, squeezable, and superficially implantable biomaterials are now underway in our laboratory.

## Discussion

This is an effective attempt to facilely construct highly conductive hydrogels with tunable mechanical properties and 3D architecture by using liquid-in-liquid 3D printing technology. Polymer inks based on aqueous PEDOT:PSS capable of simultaneously forming PEDOT:PSS colloidal particles and PDMS surfactants assemblies at the liquid-liquid

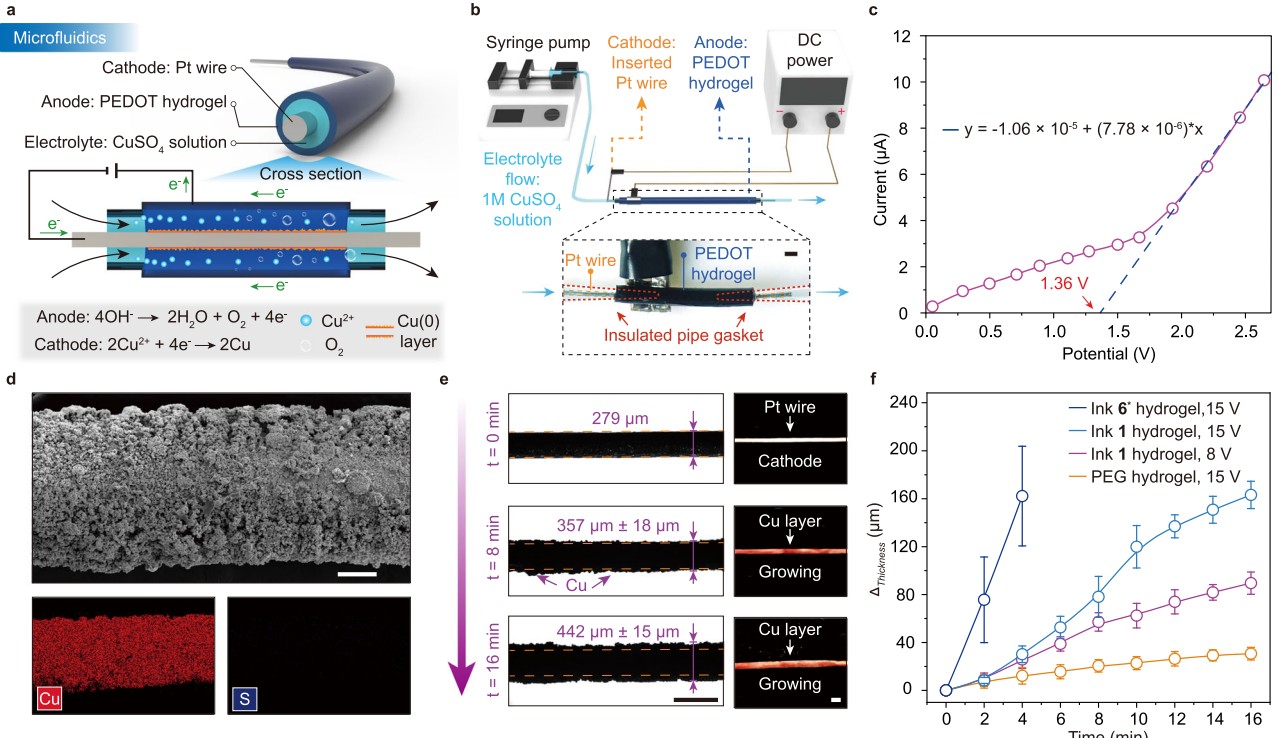

**Fig. 6 | PEDOT conductive hydrogel as an electrochemical microfluidic device.**
**a** Schematic showing the mechanism of copper electroplating inside a tubular PEDOT hydrogel. **b** Setup of the hydrogel-based electrochemical microfluidic as an electrolytic cell. The scale bar is 1 mm. **c** Determination of the decomposition potential of $CuSO_4$ aqueous solution (1 M) in the tubular PEDOT hydrogel (ink 1). **d** SEM image/EDS mapping of the Pt electrode showing the deposition of copper metal. Similar results were observed in two independent samples. The scale bar is 100 μm. **e** Evolution of the microstructure of the Pt electrode during electrolysis. Scale bars are 500 μm. **f** Copper thickness as a function of electrolysis time under different conditions and at RT. Values represent the mean, and the error bars represent the SD of the measured values ($n = 5$). The thickness variations of all Pt wires were measured after undergoing identical electrolysis processes.

interface are successfully printed into arbitrary 3D structures without additional templates or supports. Additionally, the mechanically stable interfacial assembly allows these all-liquid architectures to be extensively manipulated, such as through ionic liquid-induced gelation and UV curing. This PEDOT:PSS-based ink exhibits broad applicability for various hydrogel-forming precursors and 3D printability over a wide range of PEDOT:PSS concentrations, solution pH values and, most importantly, apparent viscosities. With this capability, we successfully printed stretchable hydrogels (subjected to >200% strain) without compromising their conductivity, integrating multiple materials into advanced 3D objects and fabricating conductive tubular hydrogels for electrosynthesis. The PEDOT gel-based NFC chips enable wireless power harvesting through biological tissues, offering a possible route for electrical signal transmission while preserving the designed 3D architecture and matched mechanical properties when applied to human tissue engineering. Although there may be a lower resolution at the current stage and the long-term stability of the electrodeposited PEDOT:PSS on bioelectrodes in a wet physiological environment needs to be further improved, the ability to spatially program the electronic conductivity, mechanical properties and material functionality of a hydrogel in three dimensions makes this liquid-in-liquid 3D printing technique highly promising for integrating conductive gels into flexible and implantable bioelectronics, directing both matter and energy across biotic-abiotic interfaces.

## Methods
### Materials
Aqueous PEDOT:PSS dispersion, CLEVIOS™ PH 1000 (1.0 - 1.3 solid content) was purchased from Heraeus and used after adjusting the solid content to 10 mg mL$^{-1}$. Aminopropyl-terminated polydimethylsiloxane (PDMS-NH$_2$, $M_n$ 4000 g mol$^{-1}$) was purchased from

Suzhou Siso new material Co.,Ltd. (China) and used as received. The ionic liquid of 1-carboxymethyl-3-methylimidazolium bis(trifluoromethylsulfonyl)imide (HOOCMIMNTF2) was kindly provided by Yu Wang lab in Sichuan University (Purity ≥ 98%). PEG-diacrylates ($M_n$ 6000 g mol$^{-1}$ and $M_n$ 20000 g mol$^{-1}$) were synthesized according to the literature[68]. Rabbit anti-F4/80 antibody (70076S)、rabbit anti-Ly6G antibody (87048S)、and rabbit anti-CD206 antibody (24595T) were purchased from Cell Signaling Technology, Inc. (CST, USA). All other chemicals were purchased from Aladdin Scientific Ltd. (China) and used without further purification.

### Synthesis of poly(ethylene glycol) diacrylate ($M_n$ 6000 g mol$^{-1}$)
The synthesis was carried out following a literature procedure[68]. In a 100 mL three-neck flask under a nitrogen atmosphere, 20 g of PEG ($M_n$ 6000 g mol$^{-1}$, 1 eq.), 695 μL of triethylamine (5 eq.), and 40 mL of anhydrous tetrahydrofuran were combined and stirred until a homogeneous mixture was obtained. The flask was then placed in an ice-water bath, and a solution of 10 mL of anhydrous tetrahydrofuran containing 400 μL of acryloyl chloride (5 eq.) was added dropwise under a nitrogen atmosphere. Once the dropwise addition was complete, the reaction mixture was allowed to react overnight at room temperature. The formed salt was removed by filtration, and the filtrate was concentrated under vacuum. The resulting residue was then precipitated in ether, yielding a white solid product with 75% yield. The synthesis of poly(ethylene glycol) diacrylate ($M_n$ 20,000 g mol$^{-1}$) followed a similar procedure as described above.

### Mice
All studies of implantation experiments in vivo were approved with the permission of the Sichuan University Ethics Committee (protocol

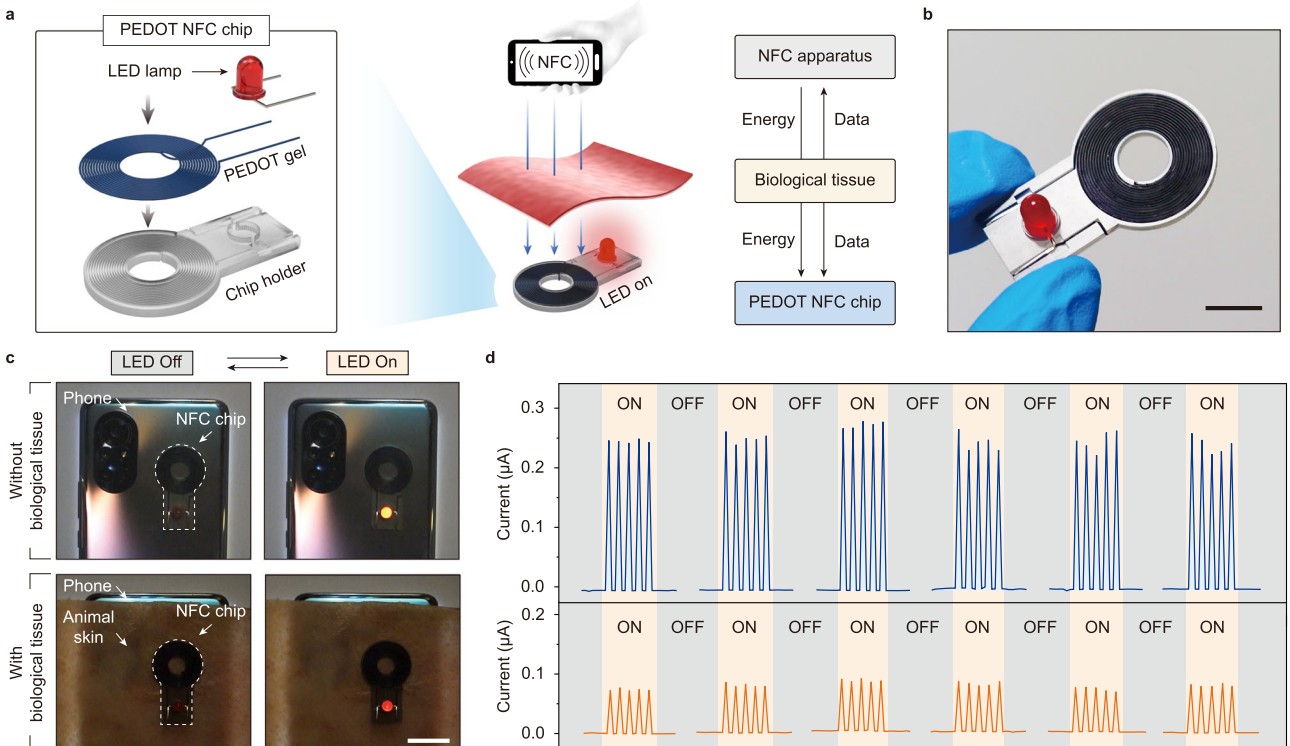

**Fig. 7 | Demonstration of PEDOT NFC devices. a** Fabrication and mechanism of printed conductive PEDOT gel that harvests power wirelessly from an NFC-enabled smartphone and lights up an LED. **b** Photograph of a PEDOT gel-based NFC chip. The scale bar is 1 cm. **c** Photograph showing that the electrical energy transmitted to the PEDOT NFC chip through a 3 mm thick animal skin is also able to light up the LED. The scale bar is 2 cm. **d** Cyclic dynamic electrical response in the NFC chip circuit.

number 2020361A) and were conducted according to the institution's guidelines. Female C57 black 6 mice (8 weeks old) were provided by the Dr. Hong Jiang laboratory (Sichuan University) and were used for all experiments in this study. The mice were housed in a controlled environment with a 12h–12h light-dark cycle (light cycle: 8:00 to 20:00), and they had access to food and water *ad libitum*. The room temperature was maintained within the range of 20–22 °C, while the humidity was kept between 30–70%, with an average daily humidity level of approximately 47%.

### 3D printing hydrogels

Water-in-oil threads were produced using a commercially available 3D printer (Ender 5 s, Creality, China), with the print heads replaced by a 30-, 25-, or 14-gauge stainless steel needle attached to a syringe pump (Longer Pump, China). To generate liquid threads with more accurate and uniform diameter, a precision dispenser (Prtronic, China) was also applied for injecting inks into liquid threads. Print head trajectories was generated using AutoCAD 2014 software, which was subsequently converted into G-code using open-source software Repetier-Host to monitor the x-y-z motion of the print head. Lateral print head velocities were set from 0.1 to 1.5 m min$^{-1}$. Aqueous dispersions were injected at 0.05–1.5 mL min$^{-1}$, depending on the desired thread thickness. All the printing procedures were implemented in the mixed silicone oil phase (viscosity varied from 10 mPa•s to 30,000 mPa•s) containing 10 vol% aminopropyl terminated polydimethylsiloxane. Unless specified otherwise, the oil phase utilized for 3D printing is silicone oil ($\eta_e$ ~ 30,000 mPa•s) that contains 10 vol% PDMS-NH$_2$.

For the hydrogels with a single chemically crosslinked network, the liquid ink threads were irradiated under UV$_{395\,nm}$ light for various time to ensure the chemical crosslinking after printing. For the hydrogels with two interpenetrating networks, freshly prepared ink containing 2 wt% ionic liquid HOOCMIMNTF2 was charged into a syringe and implemented a similar 3D printing process. Subsequently, the liquid thread was static for 10 min in the silicone oil to ensure sufficient physical gelation. After UV curing, the hydrogels were rinsed thoroughly with hexane.

### Post-treatments

Post-treatments were carried out via washing the hydrogels by using ethylene glycol for 1.5 h, and annealing at 130 °C for 10 more minutes. If needed, the dry-annealed polymer gels were further immersed in water for 24 h to convert into the hydrogel state.

### Characterization

A Discovery Hybrid Rheometer HR 20-TA instrument (USA) was used to perform tensile testing. Rheological measurements were conducted by using Anton-Paar's Modular Compact Rheometer 302 (Austria). For the physical crosslinking measurement, a 25 mm parallel plate was used and the measurements were taken at 25 °C at 1 Hz and a strain rate of 0.1%. SEM images of the hydrogels were obtained by lyophilizing the hydrogels, and images were obtained using an Apreo 2 SEM (Thermo Fisher Scientific, USA). Conductivity was calculated by measuring resistance using a Keithley 2400 two-point probe head. Reported values reflect an average over a minimum of three to five measurements obtained for each condition. The brightfield images were taken by using a Leica M205 FCA microscope (Germany) and Leica DMi 8 microscope (Germany), and the data were analyzed with FIJI software. GIWAXS data were obtained at 1W1A Diffuse X-ray Scattering Station, Beijing Synchrotron Radiation Facility (BSRF-1W1A). The monochromatic of the light source was 1.54 Å. The data were recorded by using the two-dimensional image plate detector of Eiger 2 M from Dectris, Switzerland.

## Interfacial tension (IFT) measurements

Dynamic surface tension measurements were performed by fitting the profile of a pendant drop of aqueous PEDOT:PSS dispersion, immersed in toluene containing PDMS-NH$_2$ surfactants, to the Young-Laplace equation. Droplet profiles and fitting were performed using Krüss DSA25 Drop Shape Analyzer, with surface tensions calculated using the DSA Advance software. The evolution of IFT with time was recorded after slowly injecting the highly diluted PEDOT:PSS aqueous droplets with an initial volume of 20 µL into the toluene phase.

## Copper electroplating in the conductive hydrogels

The hollow structured hydrogels with different electrical conductivities (ink 1 and ink 6*) were fabricated and soaked in 1 M CuSO$_4$ solution for 12 h before setting up. The open ends of the hydrogel tubes were sealed tightly by using two identical insulated pipe gaskets, and a clean and bare platinum wire (diameter: 0.28 mm) was implanted into the hollow hydrogels along the central axial of the hydrogel tubes. In a humidity-saturated environment, the hydrogel shell was connected with the positive electrode of a DC source, and the Pt wire was connected to the negative electrode of the DC source to complete the circuit. A steady stream of 1 M CuSO$_4$ solution was pumped through the inlet at a flow rate of 0.05 mL min$^{-1}$, and a certain voltage was applied to initiate the electrolytic reaction.

## NFC chip fabrication

A small chip holder that possessed a continuous coil groove was obtained via DLP 3D printing of commercial resin (MiiCraft Prime 110). Next, a printed PEDOT filament (ink 5*), 1 m long with a thickness of 200 µm, was strung inside the groove under room temperature. An LED was embedded in the chip to form a close circuit.

## Reporting summary

Further information on research design is available in the Nature Portfolio Reporting Summary linked to this article.

# Data availability

The data that support the plots within this paper and other finding of this study are presented in the main article and the Supplementary Materials. All other additional data are available from the corresponding author upon request. Source data are provided with this paper.

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

## Acknowledgements

The authors thank to Dr. Yu Wang in Sichuan University for help in tensile testing. The authors gratefully acknowledge the cooperation of the beamline scientists at BSRF-1W1A beamline. This work was financially supported by the Science and Technology Project of Sichuan Province (2021YFH0183, 2022ZYD0030 and 2023NSFSC0990).

## Author contributions

X.X., H.L. and W.F. conceived the idea and designed the study. X.X. and Z.X. carried out the printing experiments, mechanical characterizations and electrical conductivity measurements. X.X. performed the electrochemical tests, chip fabrication and cell experiments. X.Y and H.J performed the subcutaneous implantation and histological analysis on the mice. X.X., H.L. and W.F. analyzed and interpreted the results. X.X. and W.F. wrote the manuscript.

## Competing interests

The authors declare no competing interests.
