## [Peer Review File · Nature Communications]

Liquid-in-Liquid Printing of 3D and Mechanically Tunable Conductive HydrogelsREVIEWER COMMENTS

Reviewer #1 (Remarks to the Author):

The manuscript is a well-written and comprehensive study. The authors use a liquid-in-liquid 3D printing approach to 3D print mechanically tunable conductive hydrogel induced by an assembly at the interface. They also demonstrated some potential applicability on electrochemical microfluidic and near-field communication devices. However, the paper needs some clarity and comprehensiveness in the background description. Therefore, I would recommend being considered publishing in Nature communications if the following concerns are addressed:

Major comments:

1. Some studies out there used the same approach of having PEDOT-PSS ink printed within another support phase, but the author didn't mention them and did not include how their systems are compared to other studies. The author should explain the novelty of their work compared to other liquid-in-liquid printing of PEDOT-PSS.

-a relevant review: Martinelli, Angelo, et al. "3D printing of conductive organic polymers: challenges and opportunities towards dynamic and electrically responsive materials." *Materials Today Chemistry* 26 (2022): 101135.

2. The introduction needs to describe previous relevant studies on liquid-in-liquid 3D printing. In addition, the authors should include various material systems or mechanisms used in liquid-in-liquid 3D printing. Examples of such systems are nanoparticle surfactants and surfactant self-assembly. Among surfactant-self-assembly, the following paper is relevant and is not included:

-H. Honaryar, J. A. LaNasa, Elisabeth C. Lloyd, R. J. Hickey, and Z. Niroobakhsh, "Fabricating robust constructs with internal nanostructure via liquid-in-liquid 3d printing," *Macromolecular Rapid Communications*, 2021, 42 (22), 2100445, 2021.

Not only does the paper have similarity in the 3D printing approach, but they are also using PEGDA and chemical cross-linking, hence having the possibility to tune the mechanical properties, and need to be addressed in the relevant discussion in the paper, such as the first paragraph of page 9. Also, a recent review paper provided a comprehensive review of liquid-in-liquid 3D printing that included various material systems that are based on liquid-liquid assemblies.

-A relevant review: H. Honaryar, S. Amirfattahi, and Z. Niroobakhsh, "Associative liquid-in-liquid 3D printing techniques for freeform fabrication of soft matter", DOI:10.13140/RG.2.2.30748.97927, (2022).

3. The authors describe the aqueous phase in the introduction but provide minimal information about the oil phase. Further, in the introduction and 2.1, the assembly process for the specific system of studies can be extended; it needs to be clarified what type of process governs the interfacial activities (e.g., coagulation, coacervation, emulsifications). Other examples:

-Sun, Shuyi, Yuzheng Luo, Yang Yang, Jie Chen, Shuailong Li, Zhanpeng Wu, and Shaowei Shi.

"Supramolecular Interfaces and Reconfigurable Liquids Derived from Cucurbit [7] uril Surfactants." *Small* 18, no. 44 (2022): 2204182.

-Gratson, Gregory M., Mingjie Xu, and Jennifer A. Lewis. "Direct writing of three-dimensional webs." *nature* 428.6981 (2004): 386-386.

Further, the discussion on page 5, about very low pH could be clearer (lines 13-18). Please clarify and

use additional references to support your interpretation.

4. What do the authors mean by "effective concentration" in Line 15 of Page 3 or elsewhere throughout the manuscript?

5. Interfacial tension measurement using the conventional method for complex interfaces where the interface forms an elastic interface is questionable. The author should explain how they confirmed data acceptability and mention the measurement limitation. For reference:

-Rodríguez-Hakim, M., Anand, S., Tajuelo, J., Yao, Z., Kannan, A., & Fuller, G. G. (2020). Asphaltene-induced spontaneous emulsification: Effects of interfacial co-adsorption and viscoelasticity. *Journal of Rheology*, 64(4), 799-816.

6. What do the authors mean by "stress" in Line 34 of Page 12? Do they mean "yield stress" or "flow stress" because stress by itself is not evident in this sentence?

7. No procedure is provided for rheological measurements. What geometry or test parameters were used?

8. Page 7, line 22, can the author clarify on "skin-core" mechanism and support it from the available literature?

9. The author explained that the printability region is related to aqueous and oil-based systems concentration (Figure 2d). However, Table 1 needs to be clarified what the concentration of oil bath for each ink is. Please clarify in the text and table.

10. Page 9, lines 7-12, please clarify in the text how the concentration of monomers (and possibly other components) is changing from inks 1-5 and explain their impact on mechanical properties. Specifically, the line 7 for the sentence starting with these hydrogels...., does the author mean by decreasing the monomer concentrations? Please clarify the sentence.

11. Page 11, line 1. The author conducted a test on the flowability of the ink (SI fig 10); there is no mention of the size of the tubes or any scale bar. The capillary force may impact the results if the tube is considered a capillary tube. Please clarify.

12. The authors talk about two interpenetrating networks on several occasions, but they are not clearly described. For example, on page 11, lines 5-10, is it because of using two acrylate networks? If so, what are the primary and secondary networks? For what inks is this the case? Why they are not included in Table 1 description (only one type of PEG is used for almost all inks formulation).

13. Page 14, line 20, references to the relevant literature on electrochemical reactions or copper electroplating to be added for interested readers.

14. Section 2.6, are the hydrogels made using 3D printing or bulk for NFC chips? Since the title of the work is mainly about 3D printing, it is appropriate to describe the printing approach (if any) in this section or change the title of the paper.

Minor comments:

15. Term liquid-in-liquid 3d printing is not consistently used throughout the paper (for example, on page 6, line 18, 3d liquid-in-liquid printing). Please use the term consistently as liquid-in-liquid 3d printing.

16. "NFC" is not defined in its first instance in the abstract (Line 36).

17. It should be made clear that the range provided for the tissue robustness (Line 11, Page 2) is for the Young modulus.

18. The sentence starting with "Specifically, constructing hydrogels..." on line 34 of Page 2 is so out of context and does not fit there. No connection between the sentence before and after.

19. Formatting for the number average molecular weight should be corrected; n should be subscript (M_n), and the unit is missing.
20. Page 5, line 25, what is areal density? Is it a typo of area density?
21. Page 5, line 29, ms to be changed to milliseconds.
22. Is "Plateau-Raleigh" to be fixed to "Plateau-Rayleigh"
23. The formatting for storage and loss moduli G' and G'' (the apostrophes appear as quotation marks in the text)
24. Page 6, line 21, it is suggested that the liner be changed to lines.
25. Page 7, line 7, it is suggested that as-printed be changed to printed.
26. Page 7, table 1, typo on the bottom caption: "nozzole", to be fixed
27. The verb tense to be consistent: for example, on page 9, line 21, tended to be changed to tends
28. On page 10, line 15, the sentence has grammatical issues.
29. Page 10, line 33, ink's storage modulus to be changed to ink storage modulus
30. Page 11, line 31, what is PH1000?

Reviewer #2 (Remarks to the Author):

This manuscript reports on 3D printing of conductive PEDOT-PSS-based threads and applications to electrochemical microfluidic and NFC devices. The success relies on the interfacial assembly of PEDOT:PSS-surfactant and the formation of an elastic shell. The liquid-in-liquid printing enables the fabrication of very long conductive hydrogel threads. The authors even fabricated hollow threads for electrochemical microfluidic device. The concept is interesting and well demonstrated. But some issues need further investigations before it can be recommended for publication in Nature Communications.

1, Figure 2a is the fundamental to the concept of core-shell thread of PEDOT:PSS. However, it does not provide any useful details describing how the polymers and surfactants distribute and interact at the interface. Nor does it provide any details on the structures of the colloidal particles of PSS. Therefore, it is difficult to understand how the shell forms at the interface and why it should form a stable shell. Figure 2c show a representative core-shell structure. But no colloidal structures are observed in the shell. More SEM images of the shell should be provided to show the structure details to examine their hypothesis.

2, Soaking in ethylene glycol and dry annealing at 130°C was used to improve the modulus and conductivity of the gel. It is attributed to the formation of a dense network. In fact, this is mainly caused by water loss. Water content in the gels before and after the treatment should be measured, compared and discussed. Besides, it is no wonder that the dried gel can hardly re-swell because its the poor hydrophilicity. Moreover, the resistance slightly decreased during cyclic stretching to 50% strain. This is attributed to orientation and more π - π stacking of PEDOT chains. Can the authors use SAXS and Raman spectroscopy to examine the structural or conformation evolution of the gels?

3, The term "liquid threads (Page 13, line 9)" is misleading. Are the threads crosslinked into hydrogels?

4, What is the advantage of electroplating in the hydrogel tube? The current experimental design does

not show any advantages by doing so. It is not clear why it is necessary to do it in the tube.

Reviewer #3 (Remarks to the Author):

Paper takes well established approaches and combines into a novel way to form freestanding printed conducting polymer structures. They also show the versatility of the process in forming several configurations and performing electrosynthesis of Cu as well fabrication of a NFC. These approaches are innovative and highlights the significance of the method developed.

This paper represents a significant enhancement in the knowledge and understanding of printing 3D conducting polymer structures with multi-function a applications. The characterisation performed in detailed and comprehensive.

They mention in several places suitability for use in implantable applications however no cellular investigations were performed. Whilst PEDOT has been reported to be cytocompatibility, the fabrication approach used here may introduce some cytotoxic element.

Some suggested corrections

- Provide information on how IFT was determined. What measurements and calculations were performed?
- Sup fig 3 is 0.5mg/ml NOT 1mg/ml as indicated in the text
- In Fig 2e, it appears that at higher viscosity the fibre thickness is larger. Is this correct? if so why does this occur.
- Table 1 what oil viscosity are these conditions investigated under.

The quality of the work and the high standard of the manuscript is at a level suitable for publication in Nature Communications. I recommend publication after the authors has considered the comment regarding the cellular aspect and corrections listed above.

Reviewer #1:

The manuscript is a well-written and comprehensive study. The authors use a liquid-in-liquid 3D printing approach to 3D print mechanically tunable conductive hydrogel induced by an assembly at the interface. They also demonstrated some potential applicability on electrochemical microfluidic and near-field communication devices. However, the paper needs some clarity and comprehensiveness in the background description. Therefore, I would recommend being considered publishing in Nature communications if the following concerns are addressed:

Author Response:

We thank the reviewer for their favorable assessment of our work, and for kindly providing thoughtful feedback and suggestions to improve the manuscript quality. Below, please find our responses to each individual point raised by the reviewer:

- 1. Some studies out there used the same approach of having PEDOT-PSS ink printed within another support phase, but the author didn't mention them and did not include how their systems are compared to other studies. The author should explain the novelty of their work compared to other liquid-in-liquid printing of PEDOT-PSS. A relevant review: Martinelli, Angelo, et al. "3D printing of conductive organic polymers: challenges and opportunities towards dynamic and electrically responsive materials." *Materials Today Chemistry* 26 (2022): 101135.**

Author Response:

We thank the reviewers for the suggestion. In this study, we present a novel and straightforward approach for constructing 3D PEDOT:PSS conductive hydrogels using the interfacial jamming of PEDOT:PSS-PDMS surfactants (PPSs) at the immiscible aqueous and oil-liquid interface. To our knowledge, this is the first time that PEDOT:PSS has been printed using this method. The most significant advantage of our approach is that it eliminates the requirement for rheological properties in both ink and support bath phases, as well as strict matching of ink concentration, support bath components, and extrusion rate, which are often required in other methods such as fibrillary gelation of PEDOT:PSS by injection into coagulation baths. This is due to the fast assembly rate of the PPSs interfacial assembly process and the strong mechanical properties of the PPSs assembly layer. Our ink formulation is straightforward and clean, as surfactants or nanoparticles are not required to be mixed in aqueous ink to overcome Rayleigh-Plateau instabilities. PEDOT:PSS acts as both the jamming particle and the electrical conductive unit, greatly reducing interference factors. The robustness of this printing method makes printable PEDOT:PSS inks highly versatile, covering a wide range of PEDOT:PSS concentrations (0.1-20 mg mL⁻¹ in this study), viscosity values (the corresponding viscosity of aqueous inks ranges from 6.5-23000 mPa·s, with inks of higher viscosity also being applicable), and a broad selection of hydrogel materials. With the printed hydrogels demonstrating excellent compatibility with biological systems, our method holds enormous potential for developing implantable 3D devices that can seamlessly interact with living tissues, minimizing the risk of adverse reactions.

Revised introduction in the Main Text:

“Liquid-in-liquid 3D printing is a promising approach that allows for the freeform fabrication of soft materials.³⁵ In this technique, the ink phase is extruded into a bath phase while a translation stage moves according to a 3D design,

shaping the soft material into complex structures, while suppressing Rayleigh-Plateau instabilities and minimizing the deformation of the extruded ink in the bath phase are the primary challenges during the printing process. With the exception of embedded extrusion 3D printing, which requires specific support bath phases with shear-thinning or solid-fluid transition capabilities,^{36,37} other liquid-in-liquid printing technologies rely on self-assembly,³⁸ jamming,³⁹ and coacervation⁴⁰ to stabilize the filament extruded from the jet nozzle. These approaches partly eliminate the need for rheological properties in both ink and support bath phase. Despite these advantages, reports on the use of liquid-in-liquid 3D printing for PEDOT:PSS conductive hydrogels are still rare. While some cases of fibrillary gelation of PEDOT:PSS by injection into coagulation baths^{40, 41, 42} have been reported, the fabrication method still requires strict matching of ink concentration, support bath components, and extrusion rate due to the limited ability to overcome Rayleigh-Plateau instabilities. Although considerable efforts have been devoted to 3D printing conductive polymers, constructing 3D hydrogels with tunable mechanical properties, high conductivity, complicated structures, and a broad selection of ink materials remains challenging.”

“Importantly, the printable PEDOT:PSS inks are highly versatile, covering a broad range of PEDOT:PSS concentrations (in this study, the investigated concentration is 0.1-20 mg mL⁻¹) and viscosity values (the corresponding viscosity of aqueous inks ranges from 6.5-23000 mPa·s, with inks of higher viscosity also being applicable). Therefore, the mechanical properties of the cured hydrogels can be orthogonally controlled by tuning the ink formulation and cross-linking conditions. Given the printed hydrogels' excellent compatibility with biological systems, such superior printability and designability would allow for the fabrication of stretchable hydrogels without compromising conductivity, multimaterials integrated with 3D hydrogels, electrochemical devices and, notably, conductive hydrogels for implantable chips.”

- 2. The introduction needs to describe previous relevant studies on liquid-in-liquid 3D printing. In addition, the authors should include various material systems or mechanisms used in liquid-in-liquid 3D printing. Examples of such systems are nanoparticle surfactants and surfactant self-assembly. Among surfactant-self-assembly, the following paper is relevant and is not included: -H. Honaryar, J. A. LaNasa, Elisabeth C. Lloyd, R. J. Hickey, and Z. Niroobakhsh, "Fabricating robust constructs with internal nanostructure via liquid-in-liquid 3d printing," *Macromolecular Rapid Communications*, 2021, 42 (22), 2100445, 2021. Not only does the paper have similarity in the 3D printing approach, but they are also using PEGDA and chemical cross-linking, hence having the possibility to tune the mechanical properties, and need to be addressed in the relevant discussion in the paper, such as the first paragraph of page 9. Also, a recent review paper provided a comprehensive review of liquid-in-liquid 3D printing that included various material systems that are based on liquid-liquid assemblies. -A relevant review: H. Honaryar, S. Amirfattahi, and Z. Niroobakhsh, "Associative liquid-in-liquid 3D printing techniques for freeform fabrication of soft matter", DOI:10.13140/RG.2.2.30748.97927, (2022).**

Author Response:

As mentioned earlier, to our knowledge, this is the first time that PEDOT:PSS has been printed using the interfacial jamming of PEDOT:PSS-PDMS surfactants (PPSSs) at the immiscible aqueous and oil-liquid interface. The innovative approach of our research and how it addresses the existing challenges in the field of 3D printing are also emphasized in last response.

3. The authors describe the aqueous phase in the introduction but provide minimal information about the oil phase. Further, in the introduction and 2.1, the assembly process for the specific system of studies can be extended; it needs to be clarified what type of process governs the interfacial activities (e.g., coagulation, coacervation, emulsifications). Other examples: -Sun, Shuyi, Yuzheng Luo, Yang Yang, Jie Chen, Shuailong Li, Zhanpeng Wu, and Shaowei Shi. "Supramolecular Interfaces and Reconfigurable Liquids Derived from Cucurbit [7] uril Surfactants." *Small* 18, no. 44 (2022): 2204182. -Gratson, Gregory M., Mingjie Xu, and Jennifer A. Lewis. "Direct writing of three-dimensional webs." *nature* 428.6981 (2004): 386-386. Further, the discussion on page 5, about very low pH could be clearer (lines 13-18). Please clarify and use additional references to support your interpretation.

Author Response:

In this study, the oil phase has minimal requirements, requiring only water immiscibility and good solubility to PDMS surfactants. The viscosity and rheological properties of the oil phase have little impact on the interfacial assembly process of PEDOT:PSS. For long-term printing, the density of the oil phase can be adjusted to match that of the aqueous phase to prevent settling of the liquid phase under the influence of gravity. We have added more descriptions in the text to provide more clarity regarding the oil phase.

As for the PEDOT:PSS-surfactant assembly process, to gain insight into the interfacial assembly behavior of PEDOT:PSS and surfactants, we first examined the microstructure of PEDOT:PSS in aqueous dispersion. The prevailing model for the PEDOT:PSS dispersion is the presence of small hydrophobic PEDOT segments in close proximity to Hydrophilic PSS bundles, with these bundles forming a colloid of gel particles in water (*Nat. Commun.*, 2016, 7, 11287; *Adv. Mater.*, 2019, 31, 1806133). For the commercial PEDOT:PSS (CLEVIOS™ PH 1000) aqueous dispersion utilized in our study, DLS analysis reveals the existence of gel particles ranging from 10-400 nm in size (additional Figure S6), which aligns with previous literature reports (*ACS Appl. Mater. Interfaces* 2022, 14, 39159). Additionally, the zeta potential measurements show that the surface potential of the PEDOT:PSS nanoparticles is below -40 mV, indicating a potential for their electrostatic adsorption with positively charged surfactants (PDMS-NH₃⁺) at the liquid-liquid interface. The charged PEDOT:PSS particles in this study can only be dispersed in the aqueous phase, whereas the long hydrophobic chain of the PDMS surfactants enables them to stay in the oil phase. As a result, when the two interact at the liquid interface, electrostatic adsorption causes the jammed assembly to become immobilized at the liquid-liquid interface, arresting further shape changes in the drop and kinetically trapping the drop into a shape even far removed from equilibrium.

To demonstrate that the expected PEDOT:PSS-surfactant assembly behavior occurs at the interface, we investigated the co-assembly kinetics of PEDOT:PSS dispersion and PDMS surfactant using pendant drop tensiometry. This technique provides the time evolution of the interfacial tension (IFT) by fitting the axisymmetric profile of the droplet to the Young-Laplace equation. Additional Figure S1 and Figure 2c demonstrate that the presence of PEDOT:PSS in the aqueous phase alone does not exhibit strong interfacial activity, while the amine end-capped silicone oil in the oil phase possesses a low degree of interfacial activity (as indicated by its IFT value at its steady state). However, when both components are present in each phase simultaneously, the interfacial tension decreases dramatically and reaches a low value. Therefore, it is reasonable to conclude that the amine-capped silicone oil initially assembles at the interface in the form of PDMS-NH₃⁺, and the negatively charged PEDOT:PSS particles then quickly diffuse to the

interface and electrostatically interact with the amine end-groups on the silicone oil to form the PEDOT:PSS-surfactants assembly. This finding is also supported by the observation that deprotonation of the PDMS-NH₂ surfactant at higher pH values (>11) prevents such PPSs interfacial assembly (Figure 2b).

To visually investigate the PPSs assembly layer, we fabricated a planar layer of PPSs assembly at the liquid-liquid interface using a mold, and then collected the layer onto a glass slide. As shown in additional Figure S6, when the aqueous and oil phases are brought into contact, interfacial assembly occurs, resulting in the formation of a flat and uniform film at the interface. The assembled layer remains stable in water due to the cross-linked structure, allowing it to be easily transferred to a glass substrate. Finally, SEM images reveal the densely packed PEDOT nanoparticles at the interface layer, confirming our proposed interface assembly hypothesis.

We have expanded the main text and supplementary information to provide a more detailed and comprehensive explanation of our research findings about PPSs interfacial assembly.

Several studies have reported on the impact of pH on the nanoparticle-surfactants interfacial assembly behavior (*Adv. Mater.*, 2016, **28**, 6612; *Nano Lett.*, 2017, 17, 6453). In this revision, we conducted more experiments to measure the zeta potential of the dispersion of PEDOT:PSS at different pHs (additional Figure S3). We observed that PEDOT:PSS tended to exhibit a higher negative charge under more acidic conditions. Based on our findings, we can reasonably conclude that faster assembly occurs under more acidic conditions due to the stronger electrostatic interactions. However, at pH levels lower than 2.55, the initial adsorption of PEDOT:PSS particles to the interface occurred too rapidly, preventing the formation of regular droplet shapes for IFT measurements. When the pH was adjusted further to 1.21, the PEDOT:PSS droplet became opaque and parts of the assembly containing PEDOT:PSS were removed from the aqueous phase into oil. This observation may be due to the formation of micelles (water in oil (W/O)) at the liquid phase, resulting from the very low interfacial tension and osmotic pressure at low pH (*Angew Chem Int Ed*, 2014, **53**, 8240).

New text added in the Main Text:

“Due to the quick assembly, the oil phase, even with a low viscosity ($\eta_e \sim 10$ mPa.s), is also capable of quickly causing the PPSs to form threads from the liquid (**Figure 2e** and Supplementary Figure 7). Thus, the oil phase has minimal requirements, with only water immiscibility and good solubility to PDMS surfactants being necessary. The viscosity and rheological properties of the oil phase have little impact on the PPSs interfacial assembly process. For long-term printing, the density of the oil phase can be adjusted to match that of the aqueous phase to prevent the printed aqueous structures from settling under gravity.”

“The typical conformation of PEDOT:PSS in water is expected to be colloidal particles that consist of a hydrophilic PSS shell surrounding a hydrophobic PEDOT core region. Although PEDOT:PSS in the aqueous phase alone does not possess significant interfacial activity (Supplementary Figure 1), the addition of a PDMS surfactant terminated with primary amines (PDMS-NH₂) to an immiscible oil solution leads to the self-assembly of PEDOT:PSS colloidal particles and PDMS-NH₂ surfactants at the water-oil interface, forming a jammed PEDOT:PSS-surfactants (PPSs) composite (Figure 2a). The rapid interfacial assembly can be assumed to be a result of electrostatic interactions between the exposed sulfonate groups (R-SO₃⁻) in the PSS-rich shell and protonated amino groups (PDMS-NH₃⁺) from the oil phase at the liquid-liquid interface. Since the two components can only exist in their respective phases, strong electrostatic adsorption causes the jammed assembly to become immobilized at the liquid-liquid interface,

arresting further shape changes in the drop and kinetically trapping the drop into a shape even far removed from equilibrium.”

“To demonstrate that the expected PPSs assembly behavior occurs at the interface, we investigated the co-assembly kinetics of PEDOT:PSS dispersion and PDMS surfactant using pendant drop tensiometry. This technique provides the time evolution of the interfacial tension (IFT) by fitting the axisymmetric profile of the droplet to the Young-Laplace equation. The pH dependence of the sulfonate-ammonium ion pairing was firstly investigated by monitoring the IFT of pendant droplets of a diluted commercial PEDOT:PSS dispersion (0.50 mg mL^{-1}) in a toluene phase consisting of bis(3-aminopropyl)-terminated PDMS-NH₂ (M_n 4000 g mol^{-1} , 10 vol%) while varying the pH of the aqueous phase. We found that the rate of interfacial self-assembly of PPSs increases with decreasing pH (**Figure 2b** and Supplementary Figure 2). At higher pH values (>11), PDMS-NH₂ loses its ability to assemble at the interface with PEDOT:PSS particles due to deprotonation.”

“With further decreasing the pH until the pKa of PSS (PSS is assumed to have an average pKa of approximately 1.5⁴⁷), PEDOT:PSS tends to exhibit a higher negative charge (Supplementary Figure 3). This results in stronger electrostatic interactions between the PEDOT:PSS particles and surfactants in such pH conditions. For pH values less than 3 or for higher PEDOT:PSS concentrations, the initial adsorption of PEDOT:PSS particles to the interface at pH 2.55 is therefore too rapid to form regular droplet shapes, preventing the measurement of IFT using the shape-fitting method. More specifically, adjusting the pH to 2.07 can lead to a continuous jet without breakup when the flow rate is set to 0.07 mL min^{-1} (Supplementary Figure 4). When the pH was further adjusted to 1.21, the PEDOT:PSS droplet became opaque, and parts of the assembly containing PEDOT:PSS were removed from the aqueous phase into the oil phase. This phenomenon may be caused by the formation of micelles (water in oil (W/O)) at the liquid-liquid interface, which is attributed to the very low interfacial tension and osmotic pressure at low pH levels.^{48,49} These results indicate that there is a broad pH range from 2.07 to 10.87 that is compatible with this PEDOT:PSS-surfactant self-assembly system.”

“The formation of the PPSs interfacial assembly was ultimately confirmed through the fabrication of a planar layer of PPSs using a mold, as shown in Supplementary Figure 6. SEM images revealed the densely packed PEDOT:PSS nanoparticles at the assembly layer, confirming our proposed interface assembly hypothesis.”

New Figure Added in the SI:

Supplementary Figure 1. PEDOT:PSS-surfactants (PPSs) assembly behavior at the liquid-liquid interface. a,b The presence of PEDOT:PSS in the aqueous phase alone does not exhibit strong interfacial activity, while the amine end-capped silicone oil (PDMS-NH₂) in the oil phase possesses a low degree of interfacial activity (as indicated by its IFT value at the steady state). When both components are present in each phase simultaneously, the interfacial tension decreases dramatically and reaches a low value. It is reasonable to conclude that the amine-capped silicone oil initially assembles at the interface in the form of PDMS-NH₃⁺, and the negatively charged PEDOT:PSS particles then quickly diffuse to the interface and electrostatically interact with the amine end-groups on the silicone oil to form the PPSs interfacial assembly. **c** The droplet morphology and its buckling behavior when the PEDOT:PSS dispersion is withdrawn. Well-defined wrinkles are only observed at the interfacial area with PPSs assembly.

Revised caption in SI:

Supplementary Figure 2. pH-dependent PEDOT:PSS-PDMS surfactant interfacial assembly (PPSs). a Temporal evolution of the interfacial tension (IFT) of aqueous PEDOT:PSS dispersions (0.5 mg mL^{-1} , pH range of 3.01–11.82) introduced into solutions of PDMS-NH₂ in toluene (10 vol%). **b.** Droplet morphologies of PEDOT:PSS dispersion (0.5 mg mL^{-1} , pH 1.21–2.55) introduced into solutions of PDMS-NH₂ in toluene (10 vol%). At pH lower than 2.55, the initial adsorption of PEDOT:PSS particles to the interface is too rapid to form regular droplet shapes for IFT measurements. Adjusting the pH further to 1.21, the PEDOT:PSS droplet became opaque and parts of the assembly containing PEDOT:PSS were removing from the aqueous phase into oil. This may arise from the formation of micelles (water in oil (W/O)) at the liquid phase, due to the very low interfacial tension and osmotic pressure at low pH.^[1]

New Figure added in the SI:

Supplementary Figure 3. Variation of zeta-potential of PEDOT:PSS aqueous solution with pH.

4. What do the authors mean by "effective concentration" in Line 15 of Page 3 or elsewhere throughout the manuscript?

Author Response:

We intended to use the term "effective concentration" as the minimum concentration that can be used for printing purposes. We have revised this sentence to provide a more understandable explanation.

Revised text:

“Importantly, the printable PEDOT:PSS inks are highly versatile, covering a broad range of PEDOT:PSS concentrations (in this study, the investigated concentration is 0.1-20 mg mL⁻¹) and viscosity values (the corresponding viscosity of aqueous inks ranges from 6.5-23000 mPa·s, with inks of higher viscosity also being applicable).”

5. Interfacial tension measurement using the conventional method for complex interfaces where the interface forms an elastic interface is questionable. The author should explain how they confirmed data acceptability and mention the measurement limitation. For reference: - Rodríguez-Hakim, M., Anand, S., Tajuelo, J., Yao, Z., Kannan, A., & Fuller, G. G. (2020). Asphaltene-induced spontaneous emulsification: Effects of interfacial co-adsorption and viscoelasticity. *Journal of Rheology*, 64(4), 799-816.

Author Response:

Pendant drop tensiometry is a widely used and reliable technique for measuring the interfacial energy of liquid-liquid or liquid-air interfaces, as noted in the literature (*Journal of Rheology*, 2020, **64**, 799). The technique involves forming a pendant drop on a capillary, with the degree of deviation from a perfect spherical shape of the droplet indicating the relationship between its weight and surface tension. The interfacial tension can be obtained by iteratively fitting the drop's shape to the Young-Laplace equation, provided the density difference between the two phases is known. In our study, we utilized the Krüss DSA25 Drop Shape Analyzer to obtain droplet profiles and perform the fitting process. The surface tensions were automatically calculated using the DSA Advance software.

To satisfy the conditions required for the shape-fitting method, we injected a slowly growing aqueous droplet with an initial volume of 20 µL (which has a diameter greater than 3 mm) into the toluene phase. To ensure that the interface remains undeformed, we used a highly diluted aqueous solution with a PEDOT:PSS concentration of less than 0.5 mg mL⁻¹ to form the pendant droplet. This helped to slow the speed of PEDOT:PSS-surfactant interfacial assembly and create a regular, symmetrical, and smooth droplet shape. The high capturing speed of the DSA Advance software (frame rate more than 2000 fps) also ensured that the interface remained undisturbed during each acquisition. These conditions guaranteed the accuracy and reliability of the interfacial tension data we collected. However, for pH values less than 3 or for higher PEDOT:PSS concentrations, the droplet became irregular in shape, preventing the measurement of interfacial tension using the shape-fitting method (as shown in Supplementary Figure 2).

New text added in the Main Text:

“To demonstrate that the expected PPSs assembly behavior occurs at the interface, we investigated the co-assembly kinetics of PEDOT:PSS dispersion and PDMS surfactant using pendant drop tensiometry. This technique provides the time evolution of the interfacial tension (IFT) by fitting the axisymmetric profile of the droplet to the Young-Laplace equation.”

“For pH values less than 3 or for higher PEDOT:PSS concentrations, the initial adsorption of PEDOT:PSS particles to the interface at pH 2.55 is therefore too rapid to form regular droplet shapes, preventing the measurement of IFT using the shape-fitting method.”

“**Interfacial Tension (IFT) Measurements.** Dynamic surface tension measurements were performed by fitting the profile of a pendant drop of aqueous PEDOT:PSS dispersion, immersed in toluene containing PDMS-NH₂ surfactants, to the Young-Laplace equation. Droplet profiles and fitting were performed using Krüss DSA25 Drop Shape Analyzer, with surface tensions calculated using the DSA Advance software. The evolution of IFT with time was recorded after slowly injecting the highly diluted PEDOT:PSS aqueous droplets with an initial volume of 20 µL into the toluene phase.”

6. What do the authors mean by "stress" in Line 34 of Page 12? Do they mean "yield stress" or "flow stress" because stress by itself is not evident in this sentence?

Author Response:

Thanks for bringing this to our attention. Just to clarify, we were referring to the yield stress in our previous statement.

Revised text:

“A significant advantage of using PEDOT:PSS-surfactant films to stabilize the liquid hydrogel precursor is that this printing technique refrains from restricting the viscosity and yield stress of the printing ink.”

7. No procedure is provided for rheological measurements. What geometry or test parameters were used?

Author Response:

Thanks for bringing this to our attention. Test parameters of the rheological measurements are added in the revised text.

Revised text:

“Rheological measurements were conducted by using Anton-Paar's Modular Compact Rheometer 302 (Austria). For the physical crosslinking measurement, a 25 mm parallel plate was used and the measurements were taken at 25 °C at 1 Hz and a strain rate of 0.1%.”

8. Page 7, line 22, can the author clarify on "skin-core" mechanism and support it from the available literature?

Author Response:

The skin-core structure typically arises from differences in components within different regions, such as lower water content in the sheath of a polymer fiber (*Nat. Commun.*, 2019, **10**, 5293), or variations in processing parameters such as coagulation rate (*ACS Appl. Polym. Mater.*, 2019, **1**, 2157). In our research, we attribute the skin-core structure to differences in acrylate monomer concentration between the liquid filament's shell and core, which result from the interfacial self-assembly of PEDOT:PSS-surfactants (PPSs) during the printing process. We have two lines of evidence supporting this explanation. Firstly, the PPSs self-assembly forms a jammed and "solid-like" film at the liquid-liquid interface (additional Figure S6), resulting in a much lower concentration of PEG-based monomer in the skin film than in the inner core regions. This creates differences in morphology and mechanical properties between the skin and core after photo-induced chemical crosslinking. We observed that the PPSs assembly shell can be peeled off when the cured hydrogel is removed from viscous silicone oil ($\eta_e \sim 30000$ mPa·s), which suggests a relatively weak binding of the skin-core structure under this condition. Secondly, we performed a para-control experiment by crosslinking ink 1 in a transparent glass tube without PPSs interfacial assembly (additional Figure S9). The resulting hydrogel did not exhibit a skin-core structure, further confirming that the PPSs interface assembly is responsible for the component difference leading to the skin-core structure.

New text added in the Main Text:

"The formation of the "skin-core" structure can be attributed to the differences in PEDOT:PSS particle density and PEG monomer concentration, which result from the PPSs interfacial assembly (Supplementary Figure 9). This provides additional evidence for the liquid-liquid interfacial assembly that contributes to the hydrogel formation in desired shapes."

New Figure Added in the SI:

Supplementary Figure 9. Fabrication of PEDOT@PEG hydrogel without PPSs interfacial assembly. a Schematic representation of UV-curing of a PEDOT@PEG hydrogel (ink 1) in a glass tube. **b** Cross-sectional SEM images of the resulting PEDOT@PEG gel (ink 1) after freeze-drying. The absence of PPSs interfacial assembly in the fabrication process results in a lower PEDOT:PSS density on the gel's surface, in contrast to the PPSs assembly

layer shown in Supplementary Figure 6c. It is reasonable to attribute the skin-core structure shown in Figure 3c to differences in acrylate monomer concentration between the liquid filament's shell and core, which result from the interfacial self-assembly of PEDOT:PSS-surfactants (PPSs) during the printing process.

9. The author explained that the printability region is related to aqueous and oil-based systems concentration (Figure 2d). However, Table 1 needs to be clarified what the concentration of oil bath for each ink is. Please clarify in the text and table.

Author Response:

Thanks for pointing out this doubt. We have added the description of oil-based systems in the text and table footnote.

New text added in the Main Text:

“Unless specified otherwise, the oil phase utilized for 3D printing is silicone oil ($\eta_e \sim 30000$ mPa·s) that contains 10 vol% PDMS-NH₂.”

Revised Table 1 in the Main Text:

Formulation Number	PEDOT:PSS concentration (mg mL ⁻¹)	Additive monomer	Monomer concentration (mg mL ⁻¹)	Ionic liquid concentration (wt%)	UV _{$\lambda=395\text{nm}$} irradiation time (min)	Post-treatment (Y/N)	Modulus (kPa)	Strain at break (%)	Conductivity (S m ⁻¹)
Ink 1	5	PEG1000	385	—	5	N	3875.0	76	1.0×10^{-2}
Ink 2	5	PEG6000	455	—	5	N	195.2	186	9.2×10^{-3}
Ink 3	5	PEG20000	313	—	5	N	99.0	287	8.5×10^{-2}
Ink 4	5	PAAM/PEG20000	263/263	—	5	N	161.5	800	1.9×10^{-2}
Ink 5	9	PEG1000	100	—	2	N	41.2	136	5.6×10^{-2}
Ink 5'	9	PEG1000	100	—	2	Y	685.0	127	61.8
Ink 6	9	PEG1000	100	2	2	N	33.2	96	6.8×10^{-1}
Ink 6'	9	PEG1000	100	2	2	Y	54.4	130	302.0
Ink 7	9	PEG20000	100	2	2	N	7.3	263	2.0×10^{-1}
Ink 7'	9	PEG20000	100	2	2	Y	26.8	275	163.4

All hydrogel samples were printed using a $\Phi 1.6$ mm nozzle with 0.5 m min^{-1} print head speed and 1.0 mL min^{-1} ink flow rate.
All hydrogel samples were printed in oil phase (10 vol% PDMS-NH₂ in silicone oil, $\eta_e \sim 30000$ mPa·s).

10. Page 9, lines 7-12, please clarify in the text how the concentration of monomers (and possibly other components) is changing from inks 1-5 and explain their impact on mechanical properties. Specifically, the line 7 for the sentence starting with these hydrogels...., does the author mean by decreasing the monomer concentrations? Please clarify the sentence.

Author Response:

Thanks for pointing out this doubt. As UV curing time, monomer concentration, and the selection of hydrogel-forming precursors can significantly affect the mechanical properties of hydrogels, we conducted comparative

experiments to investigate the impact of all these factors on the mechanical properties of ink 1-5, as listed in Table 1. Figure S10 provides a detailed comparison of the effects of UV curing time and monomer concentration alone.

Revised text:

“To test the compatibility of this system, five cylindrical PEDOT:PSS hydrogels (diameter: 1.5 mm) were fabricated by printing the inks 1-5 listed in Table 1. In all cases, the interfacial assemblies reached their steady state rapidly for subsequent UV curing. The hydrogels printed with different ink formulations and curing conditions exhibit distinct transitions in mechanical characteristics, transitioning from stiff (Young’s modulus: 3875 kPa) to soft (Young’s modulus: 41.2 kPa) (Figure 3e).

11. **Page 11, line 1. The author conducted a test on the flowability of the ink (SI fig 10); there is no mention of the size of the tubes or any scale bar. The capillary force may impact the results if the tube is considered a capillary tube. Please clarify.**

Author Response:

Thanks for bringing up this concern. We have addressed it by adding a scale bar to the figure. Given that the inside diameter of the tube is 1.5 mm, the capillary force is negligible compared to the gravity of the injected ink liquid. As shown in Figure S10 (Figure S17 in the revised version), when the physical gelation time is less than 10 minutes, the ink in the tube remains in a liquid state and flows under the action of gravity, with some liquid remaining on the tube wall. However, when the gelation time is prolonged to 15 minutes, the ink appears as a bulk and loses its fluidity in the glass tube. Based on this observation, we conclude that the ink mixture can maintain its flowability for at least the first 8 minutes after adding the ionic liquid.

Revised Figure caption in the SI text:

“**Supplementary Figure 17.** Flowability of the PEDOT ink when mixed with 2 wt% ionic liquid HOOCCMIMNTF₂ (ink 6). 200 μ L fresh ink 6 was injected into a glass tube and left to sit for a specified time period. The tube was then tilted at a 45-degree angle to observe the ink's flow behavior.

Given that the inside diameter of the tube is 1.5 mm, the capillary force is negligible compared to the gravity of the injected ink liquid. When the physical gelation time is less than 10 minutes, the ink in the tube remains in a liquid state and flows under the action of gravity, with some liquid remaining on the tube wall. However, when the gelation time is prolonged to 15 minutes, the ink appears as a bulk and loses its fluidity in the glass tube. Based on this observation, we conclude that the ink mixture can maintain its flowability for at least the first 8 minutes after adding the ionic liquid.”

12. **The authors talk about two interpenetrating networks on several occasions, but they are not clearly described. For example, on page 11, lines 5-10, is it because of using two acrylate networks? If so, what are the primary and secondary networks? For what inks is this the case? Why they are not included in Table 1 description (only one type of PEG is used for almost all inks formulation).**

Author Response:

Thanks for bringing up this issue. In this manuscript, the term "double networks" refers to the combination of two distinct networks: the primary network resulting from the physical gelation of PEDOT:PSS (referred to as the PEDOT:PSS network) and the secondary network created by the chemical cross-linking of acrylate monomers.

Macroscopically connecting PEDOT:PSS particles to form a conductive network is a well-established method for increasing the number of charge conduction pathways (*Nat. Mater.*, 2013, **12**, 1038; *Adv. Mater.*, 2020, **32**, 1904752). To enhance the conductivity of PEDOT:PSS-based hydrogels, we incorporated the ionic liquid HOOCMIMNTF2 into printing inks (ink **6** and ink **7**) to trigger the physical gelation of PEDOT:PSS after printing. This results in the formation of a primary network through π - π stacking of PEDOT. The lyophilized sample of gelation of ink **6** confirms the homogeneity and porosity of the PEDOT:PSS network (Figure 4d, top image).

However, the PEDOT:PSS network is highly brittle even after gelation for 10 minutes. Therefore, we introduced a secondary acrylate network into the PEDOT:PSS network by UV curing (Figure 4a and b). The SEM image shows the interpenetrating structure of these two networks (Figure 4d, bottom image). The hydrogels with double networks generated from ink **6*** and ink **7*** exhibit considerable mechanical properties and electrical conductivity that far exceeds those of hydrogels with only a single acrylate network (Figure 4e and 4f, Table **1**).

In short, the hydrogels with double networks can be only obtained when an ionic liquid was added to the ink formulation. These double networks consist of the primary network formed by the physical gelation of PEDOT:PSS and a secondary network of acrylate networks.

We have revised this sentence to provide a clearer explanation.

Revised text:

“Although the PEDOT:PSS networks (primary network) behave similarly to solids after 10 min of gelation, they are still highly brittle if manipulated further. An acrylate network (secondary network) was then consequently introduced into the PEDOT:PSS network by UV curing for 2 min (**Figure 4d**), resulting in mechanical reinforcement of the resultant polymer matrix (**Figure 4e**). The printed PEDOT@PEG hydrogels with two interpenetrating networks could be stretched to 100% elongation (PEG1000 (ink **6**)) and even up to 270% elongation (PEG20000 (ink **7**)) prior to fracture.”

13. Page 14, line 20, references to the relevant literature on electrochemical reactions or copper electroplating to be added for interested readers.

Author Response:

We thank the reviewers for the suggestion. We have added relevant literatures to the text.

New text added in the Main Text:

“To verify the capability of the printed tubular PEDOT:PSS hydrogel in creating an electric field of coaxial cylinders,

we investigated the use of the printed 3D tube as an electro-microfluidic device⁶² for conducting electrochemical reactions, such as copper electroplating.⁶³”

- 14. Section 2.6, are the hydrogels made using 3D printing or bulk for NFC chips? Since the title of the work is mainly about 3D printing, it is appropriate to describe the printing approach (if any) in this section or change the title of the paper.**

Author Response:

Thanks for bringing up this concern. We would like to clarify that in Section 2.6, we utilized the 3D printing method described in Figure 5c to create filament-shaped PEDOT@PEG gels, which were then assembled onto a chip holder to form the NFC chip. The holder plays a role in preventing contact short circuits between the conductive PEDOT gels. The printed PEDOT coil, which is approximately 1-meter long, enables the NFC device to harvest wireless signals from a smartphone and transmit electrical energy. Moreover, as shown in additional Figure S22 to Figure S24, the PEDOT-based conductive gels exhibit excellent biocompatibility, indicating their potential application in implantable wireless sensing and electrical impulse delivery devices. We have added more detailed descriptions in the text to clarify the fabrication process of the chip.

Revised text:

“As a proof-of-concept of integration of these conductive gels into applicable devices, we fabricated a simple PEDOT-based NFC antenna at room temperature. As shown in **Figure 7a**, we used the 3D printing method detailed in Figure 5c to produce the filament-shaped PEDOT gels (ink **5***, approximately 1 meter long), which were then strung onto a chip holder that possessed a continuous coil groove to prevent contact short circuits between the conductive PEDOT gels (**Figure 7b**).”

- 15. Term liquid-in-liquid 3d printing is not consistently used throughout the paper (for example, on page 6, line 18, 3d liquid-in-liquid printing). Please use the term consistently as liquid-in-liquid 3d printing.**

Author Response:

We appreciate the reviewers for bringing this to our attention, and we have now standardized the format for liquid-in-liquid 3D printing.

- 16. "NFC" is not defined in its first instance in the abstract (Line 36).**

Author Response:

We thank the reviewers for bringing this to our attention. We have added the full name of "NFC" to the abstract section.

- 17. It should be made clear that the range provided for the tissue robustness (Line 11, Page 2) is for the Young modulus.**

Author Response:

We thank the reviewers for the suggestion. We have revised the sentence to clarify that the biomechanical properties of biological tissues are typically characterized in terms of stiffness, specifically the elastic modulus.

Revised text:

“To mimic the mechanical properties of a wide range of biological tissues from ultrasoft to stiff (with elastic modulus values ranging from 0.1–500 kPa),¹⁵ conductive hydrogels are required to form mechanically compliant interfaces without compromising their electronic performance.”

18. The sentence starting with "Specifically, constructing hydrogels...." on line 34 of Page 2 is so out of context and does not fit there. No connection between the sentence before and after.

Author Response:

We thank the reviewers for bringing this to our attention. We have made revisions to the manuscript following the suggestion.

Revised text:

“Such disconnected aggregates of the conductive polymer network result in a high number of open circuits, making both strategies suffer from low conductivity ($0.01\text{--}2.2\text{ S m}^{-1}$), even with a relatively high PEDOT:PSS content. Additionally, constructing hydrogels in 3D freeform such as overhanging structures does not accommodate soft objects in the absence of auxiliary support. While intensive efforts have been devoted to 3D printing of conductive polymers, it is still challenging to construct 3D hydrogels that possess tunable mechanical properties, high conductivity and complicated structures that are usually impossible due to gravity.”

19. Formatting for the number average molecular weight should be corrected; n should be subscript (Mn), and the unit is missing.

Author Response:

We have made revisions to the manuscript, including adding the correct symbol for number average molecular weight and specifying the unit as g mol^{-1} .

20. Page 5, line 25, what is areal density? Is it a typo of area density?

Author Response:

The areal density, which is expressed as the number of polymer chains per square nanometer at the interface, is a commonly employed definition in studies investigating the behavior of polymers at liquid interfaces. Previous studies, such as those published in *Macromolecules* (1999, **32**, 3448), *Macromolecules* (1992, **25**, 3075), and *Advanced Materials* (2019, **31**, 1806370), have also utilized this definition to characterize polymer behavior. We have revised this sentence to provide a clearer explanation.

Revised text:

“The concentrations of PEDOT:PSS in the aqueous phase influence the rate of diffusion to the interface and the areal density of polymers at the interface, which ultimately dictate the IFT and the mechanical strength of the assembled film at the steady state.”

21. Page 5, line 29, ms to be changed to milliseconds.

Author Response:

We thank the reviewers for the suggestion. Done.

Revised text:

“When increasing the PEDOT:PSS concentration to more than 1 mg mL^{-1} , the system reached its steady state on a time scale of milliseconds, as evidenced by the behavior of instantaneous particle-surfactant self-assembly and maintenance of the droplet’s original shape for at least 12 hours (Supplementary Figure 3).”

22. Is "Plateau-Raleigh" to be fixed to "Plateau-Rayleigh"?

Author Response:

We are grateful to the reviewers for bringing this to our attention. We have made the correction and replaced "Plateau-Raleigh" with the correct term "Plateau-Rayleigh".

23. The formatting for storage and loss moduli G' and G'' (the apostrophes appear as quotation marks in the text).

Author Response:

Thank reviewers. In response, we have made revisions to the format of the storage and loss moduli.

24. Page 6, line 21, it is suggested that the liner be changed to lines.

Author Response:

Thanks for pointing out this typo.

Revised text:

“By moving the print head of a commercial extrusion-based 3D printer equipped with a syringe pump, the aqueous PEDOT:PSS dispersion was injected into the surrounding oil phase containing PDMS-NH₂ surfactant along the programmed trajectories, drawing the liquid ink into linear or curved threads.”

25. Page 7, line 7, it is suggested that as-printed be changed to printed.

Author Response:

Thanks for the suggestion. Done.

26. Page 7, table 1, typo on the bottom caption: "nozzole", to be fixed.

Author Response:

Thanks for bringing this to our attention. Done.

27. The verb tense to be consistent: for example, on page 9, line 21, tended to be changed to tends.

Author Response:

Thanks for bringing this to our attention. Done.

28. On page 10, line 15, the sentence has grammatical issues.

Author Response:

Thanks for bringing this to our attention. The text has been revised.

Revised text:

“Despite the absence of a covalent bond or strong physical interaction between the PEDOT chain and PEG network, the PEDOT gradually migrates away from the hydrogel network and into the surrounding aqueous environment after a period of more than two months (Supplementary Figure 8).”

29. Page 10, line 33, ink's storage modulus to be changed to ink storage modulus.

Author Response:

Thanks. Done.

Revised text:

“Even though the storage modulus (G') of the ink exceeded the loss modulus (G'') throughout the entire process, at this ionic liquid concentration, the ink mixture could maintain its flowability in the first 8 min.”

30. Page 11, line 31, what is PH1000?

Author Response:

PH 1000 is the product code of commercial PEDOT:PSS aqueous dispersion that used in this study.

Revised text:

“Conducting polymer solutions, such as the pristine PEDOT:PSS aqueous dispersion (CLEVIOS™ PH 1000) used in this study, are typically not suitable for direct use in extrusion-based 3D printing or inkjet printing due to their fluidity.”

We thank the reviewer again for sharing the thoughtful feedback, and we hope we have addressed these concerns to the satisfaction.

Reviewer #2:

This manuscript reports on 3D printing of conductive PEDOT:PSS-based threads and applications to electrochemical microfluidic and NFC devices. The success relies on the interfacial assembly of PEDOT:PSS-surfactant and the formation of an elastic shell. The liquid-in-liquid printing enables the fabrication of very long conductive hydrogel threads. The authors even fabricated hollow threads for electrochemical microfluidic device. The concept is interesting and well demonstrated. But some issues need further investigations before it can be recommended for publication in Nature Communications.

We thank the reviewer for these favorable comments on the manuscript. Below we would like to response to each individual point raised by the reviewer:

- 1. Figure 2a is the fundamental to the concept of core-shell thread of PEDOT:PSS. However, it does not provide any useful details describing how the polymers and surfactants distribute and interact at the interface. Nor does it provide any details on the structures of the colloidal particles of PSS. Therefore, it is difficult to understand how the shell forms at the interface and why it should form a stable shell. Figure 2c show a representative core-shell structure. But no colloidal structures are observed in the shell. More SEM images of the shell should be provided to show the structure details to examine their hypothesis.**

Author Response:

Thanks for bringing this matter to our attention.

To gain insight into the interfacial assembly behavior of PEDOT:PSS and surfactants, we first examined the microstructure of PEDOT:PSS in aqueous dispersion. The prevailing model for the PEDOT:PSS dispersion is the presence of small hydrophobic PEDOT segments in close proximity to Hydrophilic PSS bundles, with these bundles forming a colloid of gel particles in water (*Nat. Commun.*, 2016, **7**, 11287; *Adv. Mater.*, 2019, **31**, 1806133). For the commercial PEDOT:PSS (CLEVIOS™ PH 1000) aqueous dispersion utilized in our study, DLS analysis reveals the existence of gel particles ranging from 10-400 nm in size (additional Figure S6), which aligns with previous literature reports (*ACS Appl. Mater. Interfaces* 2022, **14**, 39159). Additionally, the zeta potential measurements show that the surface potential of the PEDOT:PSS nanoparticles is below -40 mV, indicating a potential for their electrostatic adsorption with positively charged surfactants (PDMS-NH₃⁺) at the liquid-liquid interface. The charged PEDOT:PSS particles in this study can only be dispersed in the aqueous phase, whereas the long hydrophobic chain of the PDMS surfactants enables them to stay in the oil phase. As a result, when the two interact at the liquid interface, electrostatic adsorption causes the jammed assembly to become immobilized at the liquid-liquid interface, arresting further shape changes in the drop and kinetically trapping the drop into a shape even far removed from equilibrium.

To demonstrate that the expected PEDOT:PSS-surfactant assembly behavior occurs at the interface, we investigated the co-assembly kinetics of PEDOT:PSS dispersion and PDMS surfactant using pendant drop tensiometry. This technique provides the time evolution of the interfacial tension (IFT) by fitting the axisymmetric profile of the droplet to the Young-Laplace equation. Additional Figure S1 and Figure 2c demonstrate that the presence of PEDOT:PSS in the aqueous phase alone does not exhibit strong interfacial activity, while the amine end-capped silicone oil in the oil

phase possesses a low degree of interfacial activity (as indicated by its IFT value at its steady state). However, when both components are present in each phase simultaneously, the interfacial tension decreases dramatically and reaches a low value. Therefore, it is reasonable to conclude that the amine-capped silicone oil initially assembles at the interface in the form of PDMS-NH₃⁺, and the negatively charged PEDOT:PSS particles then quickly diffuse to the interface and electrostatically interact with the amine end-groups on the silicone oil to form the PEDOT:PSS-surfactants assembly. This finding is also supported by the observation that deprotonation of the PDMS-NH₂ surfactant at higher pH values (>11) prevents such PPSs interfacial assembly (Figure 2b). Notably, the PDMS surfactants used in this study are capped on both ends with primary amines (NH₃⁺-PDMS-NH₃⁺ at the liquid interface), which enhances the stability of the PEDOT:PSS-surfactant assemblies. This is achieved through PDMS chains bridging adjacent nanoparticles to form an effectively cross-linked, densely packed and "solid-like" interfacial assembly layer that exhibits strong mechanical strength and resistance to deformation (*Science* 2013, **342**, 460).

To visually investigate the PPSs assembly layer, we fabricated a planar layer of PPSs assembly at the liquid-liquid interface using a mold, and then collected the layer onto a glass slide. As shown in additional Figure S6, when the aqueous and oil phases are brought into contact, interfacial assembly occurs, resulting in the formation of a flat and uniform film at the interface. The assembled layer remains stable in water due to the cross-linked structure, allowing it to be easily transferred to a glass substrate. Finally, SEM images reveal the densely packed PEDOT nanoparticles at the interface layer, confirming our proposed interface assembly hypothesis.

We have expanded the main text and supplementary information to provide a more detailed and comprehensive explanation of our research findings.

New text added in the Main Text:

“The typical conformation of PEDOT:PSS in water is expected to be colloidal particles that consist of a hydrophilic PSS shell surrounding a hydrophobic PEDOT core region. Although PEDOT:PSS in the aqueous phase alone does not possess significant interfacial activity (Supplementary Figure 1), the addition of a PDMS surfactant terminated with primary amines (PDMS-NH₂) to an immiscible oil solution leads to the self-assembly of PEDOT:PSS colloidal particles and PDMS-NH₂ surfactants at the water-oil interface, forming a jammed PEDOT:PSS-surfactants (PPSs) composite (Figure 2a). The rapid interfacial assembly can be assumed to be a result of electrostatic interactions between the exposed sulfonate groups (R-SO₃⁻) in the PSS-rich shell and protonated amino groups (PDMS-NH₃⁺) from the oil phase at the liquid-liquid interface. Since the two components can only exist in their respective phases, strong electrostatic adsorption causes the jammed assembly to become immobilized at the liquid-liquid interface, arresting further shape changes in the drop and kinetically trapping the drop into a shape even far removed from equilibrium.”

“To demonstrate that the expected PPSs assembly behavior occurs at the interface, we investigated the co-assembly kinetics of PEDOT:PSS dispersion and PDMS surfactant using pendant drop tensiometry. This technique provides the time evolution of the interfacial tension (IFT) by fitting the axisymmetric profile of the droplet to the Young-Laplace equation. The pH dependence of the sulfonate-ammonium ion pairing was firstly investigated by monitoring the IFT of pendant droplets of a diluted commercial PEDOT:PSS dispersion (0.50 mg mL⁻¹) in a toluene phase consisting of bis(3-aminopropyl)-terminated PDMS-NH₂ (*M_n* 4000 g mol⁻¹, 10 vol%) while varying the pH of the aqueous phase. We found that the rate of interfacial self-assembly of PPSs increases with decreasing pH (Figure 2b and Supplementary Figure 2). At higher pH values (>11), PDMS-NH₂ loses its ability to assemble at the interface

with PEDOT:PSS particles due to deprotonation.”

“The formation of the PPSs interfacial assembly was ultimately confirmed through the fabrication of a planar layer of PPSs using a mold, as shown in Supplementary Figure 6. SEM images revealed the densely packed PEDOT:PSS nanoparticles at the assembly layer, confirming our proposed interface assembly hypothesis.”

New Figure Added in the SI:

Supplementary Figure 1. PEDOT:PSS-surfactants (PPSs) assembly behavior at the liquid-liquid interface. a,b

The presence of PEDOT:PSS in the aqueous phase alone does not exhibit strong interfacial activity, while the amine end-capped silicone oil (PDMS-NH₂) in the oil phase possesses a low degree of interfacial activity (as indicated by its IFT value at the steady state). When both components are present in each phase simultaneously, the interfacial tension decreases dramatically and reaches a low value. It is reasonable to conclude that the amine-capped silicone oil initially assembles at the interface in the form of PDMS-NH₃⁺, and the negatively charged PEDOT:PSS particles then quickly diffuse to the interface and electrostatically interact with the amine end-groups on the silicone oil to form the PPSs interfacial assembly. c The droplet morphology and its buckling behavior when the PEDOT:PSS dispersion is withdrawn. Well-defined wrinkles are only observed at the interfacial area with PPSs assembly.

Supplementary Figure 6. Visualizing the PPSs interfacial assembly. **a** Formation of a planar layer of PPSs at the liquid-liquid interface in a mold. When the aqueous and oil phases are brought into contact, interfacial assembly occurs, resulting in the formation of a flat and uniform PPSs film at the interface. The assembled layer remains stable in water due to the cross-linked structure, allowing it to be easily transferred to a glass substrate. **b** PPSs assembly layer collected on glass substrate. **c** SEM images showing the densely packed PEDOT:PSS nanoparticles at the assembly layer. **d** Size distribution of PEDOT:PSS particles by intensity, measured at \sim pH 3.5 using dynamic light scattering (DLS).

Supplementary Figure 8. **a** The quick PPSs assembly at the liquid-liquid interface enables printing of liquid threads in low-viscous oil ($\eta_e \sim 10$ mPa·s). **b** The PPSs interfacial assembly showing superior mechanical robustness. The PPSs assembly exhibits strong mechanical strength and deformation resistance, which is achieved through the PDMS chains (in this study PDMS surfactants are capped on both ends with primary amines, i.e., NH_3^+ -PDMS- NH_3^+ at the liquid interface) that bridge adjacent PEDOT:PSS nanoparticles to form an effectively cross-linked, densely packed, and solid-like interfacial assembly layer.²

2. **Soaking in ethylene glycol and dry annealing at 130°C was used to improve the modulus and conductivity of the gel. It is attributed to the formation of a dense network. In fact, this is mainly caused by water loss. Water content in the gels before and after the treatment should be measured, compared and discussed. Besides, it is no wonder that the dried gel can hardly re-swallow because its the poor hydrophilicity.**

Author Response:

Thank the reviewers for bringing this to our attention. In our study, we aimed to enhance the electrical properties of the printed PEDOT:PSS hydrogels. To achieve this, we performed a post-treatment process by soaking the hydrogels in ethylene glycol (EG) and dry annealing them at 130 °C once the printed inks had solidified. Our experimental results (shown in Figure S12) demonstrated that both EG-treatment and annealing-treatment improved the conductivity of the gels, which is consistent with previous literature (*Advanced Electronic Materials* 2015, **1**, 1500017; *Organic Electronics* 2009, **10**, 61). Additionally, Figure S11 displays the weight loss and diameter change of the printed gels during post-treatments, while the water content values can be calculated by dividing the difference between the dried (W_{dry}) and swollen (W_{swollen}) weights by the swollen state weight, i.e., $\text{water content} = (W_{\text{swollen}} - W_{\text{dry}}) / W_{\text{swollen}}$.

We agree with the reviewers' observation that water content significantly affects the conductivity of printed gels, as water hinders the formation of internal conductive pathways in hydrogels (*ACS Appl. Mater. Interfaces* 2020, **12**, 27518). In our study, we found that the post-treatments, particularly the dry annealing process, caused a reconfiguration of the polymer morphology, leading to changes in its swelling ability and electrical conductivity. Specifically, SEM images showed that the drying of the hydrogels during the annealing treatment resulted in the collapse of the conducting polymer network into aggregates. This collapse would encourage more π - π stacking interactions between PEDOT:PSS, leading to the formation of more conductive pathways (*Nat. Commun.* 2018, **9**, 2740). To investigate whether water loss or structural change plays a more dominant role in enhancing the conductivity, we performed three different drying treatments on printed ink 5 hydrogels: lyophilization, lyophilization followed by annealing, and direct dry annealing. While the amount of water loss was almost the same for all treatments, the morphology of the treated gels differed significantly (additional Figure S13). The direct annealed sample showed a higher degree of network stacking and aggregation, leading to a conductivity that was 10 times higher than the lyophilized sample and 6 times higher than the lyophilized/annealed sample. These findings suggest that the water-loss-induced collapse of the conducting PEDOT@PEG polymer network into aggregates in the dry annealing process is the main contributor to the enhanced conductivity of the printed gels.

New text added in the Main Text:

“Besides the known mechanisms of phase separation of PEDOT and PSS during the soaking process to form a denser PEDOT network and the recrystallization of PEDOT-rich domains during dry annealing to generate higher conductive pathways, the water-loss-induced collapse of the conducting PEDOT@PEG polymer network into aggregates during the dry annealing process is another significant factor that contributes to the enhanced conductivity of the printed gels, as revealed by the comparison of different drying treatments on printed hydrogels (Supplementary Figure 13).”

Revised Figure in the SI:

Supplementary Figure 12. Mass (a), diameter (b), electrical conductivity (c) and polymer morphology (d) changes of the printed hydrogels during post-treatment process. The water content values in the hydrogels can be calculated by dividing the difference between the dried (W_{dry}) and swollen ($W_{swollen}$) weights by the swollen state weight, i.e., $water\ content = (W_{swollen} - W_{dry}) / W_{swollen}$. Such reconfiguration of the polymer morphology in post-treatment changed the swelling ability of polymers as well. The as-synthesized gel from ink 5 swelled to approximately 2.6 times its original weight, whereas the post-treated ink 5* gel underwent a reswelling process in water with only 85% reswollen weight.

New Figure Added in the SI:

Supplementary Figure 13. Comparison of different drying treatments on printed hydrogels (ink 5): lyophilization, lyophilization followed by annealing, and direct dry annealing. **a** Polymer morphology changes of the printed hydrogels (ink 5) after drying treatments. To eliminate other influencing factors, the printed hydrogel samples were directly dried without removing the uncured monomers. While some degree of pore collapse is observed for all three drying treatments, the directly dry-annealed samples show a higher degree of network stacking and aggregation, which is also illustrated in the diameter variation of the samples. **b** Changes in mass, diameter, and electrical conductivity of the printed hydrogels during different drying treatments. While the amount of water loss is almost the same for all treatments, it's reasonable to conclude that the water-loss-induced collapse of the conducting PEDOT@PEG network into aggregates in the dry annealing process is the main contributor to the enhanced conductivity of the printed gels.

Moreover, the resistance slightly decreased during cyclic stretching to 50% strain. This is attributed to orientation and more π - π stacking of PEDOT chains. Can the authors use SAXS and Raman spectroscopy to examine the structural or conformation evolution of the gels?

Author Response:

Thanks for the suggestion. To further explain this phenomenon, we conducted additional experiments to examine the conformational changes of the PEDOT@PEG gels before and after cyclic stretching. Grazing-incidence wide-angle X-ray scattering (GIWAXS) measurements (additional Figure S14) were performed to investigate the ordering in the PEDOT:PSS-based gels. Unlike pure PEDOT:PSS thin films with post-treatments (*Adv. Mater.* 2015, **27**, 3391; *Sci. Adv.* 2017, **3**, e1602076), our printed gels did not show obvious reflection characteristic of the π - π stacking of PEDOT molecules at $q \sim 1.75 \text{ \AA}^{-1}$ and PSS stacking peaks at $q \sim 1.23 \text{ \AA}^{-1}$ under the test conditions. This similarity

to the pristine PEDOT:PSS material characteristic is mainly due to the fact that there is not a significant phase separation and crystallization of PEDOT and PSS in the printed PEDOT@PEG gels. To investigate the molecular orientation, sector integrals from horizontal (0° – 30°) to vertical (60° – 90°) direction were calculated for the printed PEDOT@PEG gel (ink 5*) before and after 200 cyclic stretching to 50% strain. As scattering from the π – π stacking of the PEDOT:PSS molecules in horizontal and vertical directions characterizes the edge-on and face-on orientations of the molecules relative to the stretching direction, respectively, we found that during stretching, the PEDOT:PSS crystallites changed from a disordered arrangement to a preferred face-on molecular orientation. This was evidenced by the noticeable increment of the face-on to edge-on integration ratio.

In the same samples, we observed that $C_\alpha=C_\beta$ vibration peaks ($\sim 1438\text{ cm}^{-1}$) in Raman spectroscopy red-shifted to 1426 cm^{-1} in the PEDOT:PSS gel after stretching treatment (additional Figure S14). This shift indicates a higher proportion of the benzoid moieties in PEDOT have been converted to the quinoid structure via oxidative charge transfer doping, as previously reported (*Macromolecules* 1999, **32**, 6807; *Sci. Adv.*, 2017, **3**, e1602076; *Adv. Sustainable Syst.*, 2018, **2**, 1800085). Such conversion leads to a more planar backbone, contributing to more efficient charge delocalization and higher packing order (*Adv. Funct. Mater.*, 2005, **15**, 203), which is consistent with the increase in electrical conductivity (Figure 3g) and face-on molecular orientation observed along this direction after cyclic stretching.

New text added in the Main Text:

“Interestingly, we found that the resistance of the post-treated gels made from ink 5* gradually decreased during the stretching-releasing process (**Figure 3g**). Reversible stretching of the gel between 0 and 50% strain is likely to encourage a better orientation and more π – π stacking interactions between the PEDOT networks, which contribute to lowering the gel’s resistance by 12% after 30 cycles. Supporting this finding, grazing-incidence wide-angle X-ray scattering (GIWAXS) measurements show that the PEDOT:PSS crystallites in the gel changed from a disordered arrangement to a preferred face-on molecular orientation relative to the stretching direction during stretching. Furthermore, a red shift of the $C_\alpha=C_\beta$ vibration peak indicates that the cyclic stretching process leads to a more planar backbone of PEDOT, which contributes to more efficient charge delocalization and a higher packing order (Supplementary Figure 14).”

New Figure Added in the SI:

Supplementary Figure 14. Conformational changes of the PEDOT@PEG gels after cyclic stretching. **a** Schematic illustration of molecule orientations in edge-on and face-on configurations, with respect to the direction of polymer gel stretching. **b** 2D GIWAXS measurements of printed PEDOT@PEG gels (ink 5*) before and after reversible stretching of the gel between 0 and 50% strain. The printed PEDOT@PEG gels exhibit the similar characteristic peak to the pristine PEDOT:PSS material. **b** Ascription of face-on and edge-on regions by the χ angle: edge-on region ($0^\circ < \chi < 30^\circ$), face-on region ($60^\circ < \chi < 90^\circ$). **c** Intensity ratio change of face-on to edge-on orientations. To investigate the molecular orientation, sector integrals from horizontal (0° - 30°) to vertical (60° - 90°) direction were calculated for the printed PEDOT@PEG gel before and after cyclic stretching to 50% strain. As scattering from the π - π stacking of the PEDOT:PSS molecules in horizontal and vertical directions characterizes the edge-on and face-on orientations of the molecules relative to the stretching direction, respectively, the PEDOT:PSS crystallites changed from a disordered arrangement to a preferred face-on molecular orientation during stretching, as evidenced by the noticeable increment of the face-on to edge-on integration ratio. **d** Raman spectra illustrating the $C_\alpha=C_\beta$ vibration peak shift for the PEDOT@PEG gels (ink 5*) after reversible stretching treatment. This shift indicates a higher proportion of the benzoid moieties in PEDOT have been converted to the quinoid structure via oxidative charge transfer doping, and such conversion leads to a more planar backbone, contributing to more efficient charge delocalization and higher packing order.^{3,4}

3. The term "liquid threads (Page 13, line 9)" is misleading. Are the threads crosslinked into

hydrogels?

Author Response:

Thank the reviewers for bringing this to our attention. In this study, we use the term "liquid threads" to describe the ink filaments that have been printed but have not undergone physical gelation or chemical curing. We have added additional descriptions of this terminology in the text to clarify any misunderstandings.

New text added in the Main Text and Figure captions:

“By moving the print head of a commercial extrusion-based 3D printer equipped with a syringe pump, the aqueous PEDOT:PSS dispersion was injected into the surrounding oil phase containing PDMS-NH₂ surfactant along the programmed trajectories, drawing the liquid ink into linear or curved threads (in this study, the term 'threads' refers to liquid inks with a thread-like shape and lacking any crosslinked structure).”

“Figure 2f. The mechanically robust PPSs assembly allows the liquid thread of pure PEDOT:PSS dispersion to hang in air.”

“Figure 3b. Linear thread of printed ink **1** in silicone oils before UV curing into hydrogel.”

4. What is the advantage of electroplating in the hydrogel tube? The current experimental design does not show any advantages by doing so. It is not clear why it is necessary to do it in the tube.

Author Response:

In this study, we not only demonstrated the capability to print solid conductive PEDOT:PSS gels via single-layer PEDOT:PSS-surfactants (PPSs) interfacial self-assembly, but we also successfully printed tubular conductive gels with concentric circle structures using double-layer PPSs assembly. These findings present a promising and novel approach to producing coaxial cylindrical conductors using organic materials, which could provide advantages over traditional metal tubes in terms of flexibility and ease of production. To confirm the ability of the printed tubular PEDOT:PSS gel to create an electric field of coaxial cylinders for electrochemical reactions inside the tube, we performed a copper electroplating test reaction. Although the conductivity of PEDOT:PSS tubular gels is inferior to that of metal tubes, their ease of printing and adjustable mechanical properties make them a promising candidate for producing coaxial cylindrical conductors, including customized coaxial cables for implantable patch antennas (*IEEE Antennas and Propagation Magazine*, 2012, **54**, 210), in a more efficient and tailored way. Further descriptions of the advantages of this technique have been included in the text.

New text added in the Main Text:

“To verify the capability of the printed tubular PEDOT:PSS hydrogel in creating an electric field of coaxial cylinders, we investigated the use of the printed 3D tube as an electro-microfluidic device for conducting electrochemical reactions, such as copper electroplating.”

“The uniform copper coating on the surface of the platinum wire confirms the presence of an electric field created by coaxial cylinders.”

“Although the conductivity of PEDOT:PSS tubular gels is inferior to that of metal tubes, their ease of printing and adjustable mechanical properties make them a promising candidate for producing coaxial cylindrical conductors, including customized coaxial cables for implantable patch antennas, in a more efficient and tailored way.”

We thank the reviewer again for sharing the thoughtful feedback, and we hope we have addressed these concerns to the satisfaction.

Reviewer #3:

Paper takes well established approaches and combines into a novel way to form freestanding printed conducting polymer structures. They also show the versatility of the process in forming several configurations and performing electrosynthesis of Cu as well fabrication of a NFC. These approaches are innovative and highlights the significance of the method developed.

This paper represents a significant enhancement in the knowledge and understanding of printing 3D conducting polymer structures with multi-functional applications. The characterization performed in detailed and comprehensive.

They mention in several places suitability for use in implantable applications however no cellular investigations were performed. Whilst PEDOT has been reported to be cytocompatibility, the fabrication approach used here may introduce some cytotoxic element.

Author Response:

We thank the reviewer for these favorable comments on the manuscript.

To validate the biocompatibility of our printed PEDOT:PSS hydrogels, we performed additional experiments to assess their cytocompatibility *in vitro* and histological analysis *in vivo*, which are crucial for their potential use as implantable materials.

In order to assess the suitability of different printed PEDOT:PSS hydrogels for the survival and growth of immortalized human embryonic kidney cells (HEK 293T cells), we cultured the cells in 96-well cell culture plates and quantitatively measured their viability and proliferation (additional Figure S22). The cell viability and proliferation were evaluated using the CCK8 assay at the end of 24 h, 48 h, and 72 h of incubation. We found that cell survival with a high viability percentage of approximately 95% of all hydrogel samples, even after three days of culture. Furthermore, all samples exhibited increased cell proliferation by the end of day 3. The cell viability results exceed the ISO standard for cytotoxicity of materials, which requires a minimum viability of 70% (Standardization, I. O. F. International Standard ISO 10993–10991 Biological Evaluation of Medical Devices–Part 1: Evaluation and Testing Within a Risk Management Process (2009); *Scientific Reports*, 2022, **12**, 8259).

The anti-adhesive effect of artificial polymers on cells and tissues is crucial for maintaining good health and protecting the body from trauma and foreign objects. It is also a key parameter in the development of polymeric biomaterials for biomedical applications (*Progress in Polymer Science*, 2015, **44**, 28). PEG is well known for its bio-inertness, which suppresses non-specific protein adsorption and prevents cell adhesion. In order to determine whether the PEDOT:PSS-surfactants (PPSs) interfacial self-assembly alters the anti-adhesion properties of PEG-based gels, we conducted a set of comparative experiments (see additional Figure S23). Specifically, we investigated the culture of HEK 293T cells on the surfaces of chemically cross-linked hydrogels with and without the PPSs assembly. As expected, the directly UV-cured PEDOT:PSS hydrogels (ink 5, ink 5*, ink 6, and ink 6*, with 100 mg mL⁻¹ PEG1000 as the monomer) exhibited excellent resistance to HEK 293T cell adhesion, as demonstrated by the non-spreading of cells on the hydrogel surfaces. Surprisingly, even though the PPSs assembly process at the liquid interface alters the interfacial density of PEDOT:PSS particles as well as the concentration of PEG monomers, the

PEDOT@PEG hydrogels with PPSs assembly layer in comparative experiments also exhibited similar anti-cell adhesion properties. This anti-adhesion property is critical for preventing implant materials from failing due to unpredictable protein adsorption and cell interaction.

We evaluated the foreign body reaction of immunocompetent mice to the printed PEDOT:PSS hydrogels by implanting subcutaneously the printed PEDOT@PEG hydrogels (ink 6*, which contains the most components) in groups of three mice (additional Figure S24). Samples for histological analysis were prepared 7 days after implantation. No degradation of hydrogels was observed at this time point. Analysis of specific tissue response revealed the accumulation of pan-macrophages (F4/80) around the hybrid PEDOT:PSS hydrogel, which displayed alternative activation (CD206, > 66%) and, therefore, acted in an anti-inflammatory manner and supported tissue repair. The printed PEDOT@PEG hydrogels exhibited high biocompatibility in immunocompetent mice at this time point. Unsacrificed mice remained alive and healthy without any abnormalities for two months after subcutaneous implantation. We are continuing to monitor the foreign body reaction in the mice for a longer period.

New text added in the Main Text:

“To validate the biocompatibility of our printed PEDOT:PSS-based hydrogels for potential use as implantable materials, we performed experiments to assess their cytocompatibility in vitro and histological analysis in vivo. We found that after 72 hours of culture, the tested hydrogel samples (ink 5, ink 5*, ink 6, and ink 6*) demonstrated high cell proliferation with a viability percentage of approximately 95% (Supplementary Figure 22). Despite the changes in interfacial density of PEDOT:PSS particles and concentration of PEG monomers resulting from the PPSs interfacial assembly process, the printed PEDOT@PEG hydrogels exhibited anti-cell adhesion properties as well (Supplementary Figure 23), which are critical for preventing implant materials from failing due to unpredictable protein adsorption and cell interaction. In addition, we observed high biocompatibility of the printed PEDOT@PEG hydrogels (ink 6* with the most components) in immunocompetent mice at a time point of 7 days (Supplementary Figure 24), with unsacrificed mice remaining alive and healthy without any abnormalities for two months after subcutaneous implantation.”

New Figure added in the SI:

Supplementary Figure 22. *In vitro* cytocompatibility tests of immortalized human embryonic kidney cells (HEK 293T cells) with printed PEDOT:PSS hydrogels. **a** Schematic setup of the cell culture in 96-well plates with various printed PEDOT:PSS hydrogels. **b** Cell viability measured after 24, 48, and 72 hours of culture using Cell Counting Kit 8 (CCK8) assay. Control cells were cultured in a medium without hydrogels. **c** Live/Dead staining of HEK 293T cells at 24 and 72 hours for cell viability and proliferation evaluation. At the end of the 72-hours culture period, the cell viability percentage was almost equal (~95%) across all hydrogel samples. The increase in cell proliferation during cell culture for all the samples also demonstrates the well cytocompatibility of the printed PEDOT-based hydrogels.

Supplementary Figure 23. Evaluation of the anti-adhesive effect of PEDOT@PEG hydrogels (ink 5, ink 5*, ink 6 and ink 6* formulas) with and without PPSs interfacial assembly on HEK 293T cells. **a** Schematic showing the fabrication of PEDOT@PEG hydrogels without PPSs interfacial assembly in a mold and subsequent cell seeding on hydrogel surfaces. **b** Morphology of HEK 293T cells in direct contact with PEDOT@PEG hydrogels without PPSs interfacial assembly after 72 hours of culture time. **c** Schematic showing the fabrication of PEDOT@PEG hydrogels with PPSs interfacial assembly in a mold and subsequent cell seeding on hydrogel surfaces. **d** Morphology of HEK 293T cells in direct contact with PEDOT@PEG hydrogels with PPSs interfacial assembly after 72 hours of culture time. At the end of the 72-hours culture, the difference in cell spreading between the hydrogel and the surrounding petri dish demonstrates that both PEDOT@PEG hydrogels with and without PPSs interfacial assembly possess anti-adhesive effects on cells.

Supplementary Figure 24. *In vivo* assessment of hydrogel biocompatibility in immunocompetent mice. **a,b** Schematic diagram and photograph depicting the subcutaneous implantation of printed hydrogels (ink 6*), which is the formula that contains the most components. **c** Pathological study through H&E staining on the implantation site 7 days after surgery. **d-f** Representative immunohistochemistry (IHC) staining of markers for neutrophils (LY6G), pan-macrophages (F4/80), and M2 macrophages (CD206), respectively. The blue line indicates the hydrogel-tissue interface. The ratio of M2 macrophages to pan-macrophages (> 66%) around the hybrid PEDOT@PEG hydrogel suggests that the macrophages mainly display alternative activation and, therefore, exhibit an anti-inflammatory response, supporting tissue repair. The printed PEDOT@PEG hydrogels exhibit high biocompatibility in immunocompetent mice at this time point, as unsacrificed mice remained alive and healthy without any abnormalities for two months after subcutaneous implantation. Longer-term monitoring of the foreign body reaction in the mice is still ongoing.

Some suggested corrections:

- Provide information on how IFT was determined. What measurements and calculations were performed?

Author Response:

Thanks for the suggestion. The interfacial tension (IFT) between two immiscible liquid phases results in an increased pressure inside a pendant droplet. The correlation between the pressure difference (Δp), the radii of curvature of the surface (R_1 and R_2), and the interfacial tension is described by the Young-Laplace equation:

$$\Delta p = \sigma \cdot \left(\frac{1}{R_1} + \frac{1}{R_2} \right)$$

The degree of deviation from the spherical shape of the pendant droplet gives the relationship between the weight of the drop and its surface tension. If the density difference between the two phases is known, the surface tension can be calculated from the shape of the drop with actual dimensions. In our study, we used the Krüss DSA25 Drop Shape Analyzer to obtain droplet profiles and perform fitting, and the surface tensions were automatically calculated using the DSA Advance software.

More descriptions are added in the text.

New text added in the Main Text:

“To demonstrate that the expected PPSs assembly behavior occurs at the interface, we investigated the co-assembly kinetics of PEDOT:PSS dispersion and PDMS surfactant using pendant drop tensiometry. This technique provides the time evolution of the interfacial tension (IFT) by fitting the axisymmetric profile of the droplet to the Young-Laplace equation.”

“**Interfacial Tension (IFT) Measurements.** Dynamic surface tension measurements were performed by fitting the profile of a pendant drop of aqueous PEDOT:PSS dispersion, immersed in toluene containing PDMS-NH₂ surfactants, to the Young-Laplace equation. Droplet profiles and fitting were performed using Krüss DSA25 Drop Shape Analyzer, with surface tensions calculated using the DSA Advance software. The evolution of IFT with time was recorded after the aqueous droplet with an initial volume of 20 μ L was slowly injected into the toluene phase.”

- Sup fig 3 is 0.5mg/ml NOT 1mg/ml as indicated in the text.

Author Response:

Thank the reviewers for bringing this to our attention. We revised the caption of this Figure to make the description clearer.

Revised caption in the Figure:

Supplementary Figure 5. The ability of instantaneous interfacial PPSs to maintain the original shape of droplets or liquid threads. **a** A pendent PEDOT:PSS droplet (1 mg mL^{-1} aqueous dispersion) in toluene solution of PDMS-NH₂ surfactants (10 vol%) reaching its steady state within milliseconds and maintaining this state for at least 12 hours. **b** A printed PEDOT:PSS thread (0.5 mg mL^{-1} aqueous dispersion) in oil phase (10 vol% PDMS-NH₂ in silicone oil, $\eta_e \sim 30000 \text{ mPa}\cdot\text{s}$) maintaining its shape for 12 hours.

• In Fig 2e, it appears that at higher viscosity the fiber thickness is larger. Is this correct? if so, why does this occur.

Author Response:

Thank the reviewers for bringing this to our attention. We conducted further tests to investigate the impact of oil viscosity on the thickness of the printed threads (see additional Figure S7). We found that the diameter (d) of the printed threads in a cylindrical shape is only dependent on the print head speeds (v) and ink volumetric flow rate (Q), and not on the oil viscosity. The thickness of the threads can be calculated using the following formula:

$$d = 2 \sqrt{\frac{Q}{\pi v}}$$

Figure 2e is revised and more descriptions are added in the Supporting Information.

New Figure Added in the SI:

Supplementary Figure 7. Diameter control of printing pure PEDOT:PSS threads in oil phase with various viscosities. The PEDOT:PSS aqueous dispersion (10 mg mL^{-1}) was injected in various silicone oil containing 10 vol% PDMS-NH₂ surfactant. The diameter of the printed PEDOT:PSS threads in cylindrical shape is only related to print head speeds (v) and ink volumetric flow rate (Q) but not to the oil viscosity. The threads thickness can also be calculated by the following formula:

$$d = 2 \sqrt{\frac{Q}{\pi v}}$$

Revised Figure 1 in the Main Text:

Figure 2. PPSs interfacial assemblies for printing of a PEDOT:PSS aqueous dispersion in oil. a) Schematic of the interfacial assembly of PPSs and the elastic films they form. b) Equilibrium interfacial tension (IFT) versus pH showing the pH-dependent assembly of PPSs at the water/toluene interface, as well as the droplet morphology and its buckling behavior when the PEDOT:PSS dispersion was withdrawn. c) Temporal evolution of IFT of PEDOT:PSS aqueous dispersions (0, 0.17, 0.25 and 0.5 mg mL⁻¹) introduced to solutions of PDMS-NH₂ in toluene (10 vol%), illustrating control over the rate of PPSs assembly at the liquid interface. d) Suitable concentration profiles of PEDOT:PSS aqueous dispersion and PDMS-NH₂ surfactant for liquid-in-liquid printing, showing the wide range of printing conditions. The scale bar is 500 μm. e) Printing the PEDOT:PSS aqueous dispersion (10 mg mL⁻¹) in silicone oil with various viscosities. The scale bar is 2 mm. f) The mechanically robust PPSs assembly allows the liquid thread of pure PEDOT:PSS dispersion to hang in air. The scale bar is 1 cm.

• Table 1 what oil viscosity are these conditions investigated under.

Author Response:

Thanks for pointing out this doubt. We have added the description of oil-based systems to the text and table footnote.

New text added in the Main Text:

“Unless specified otherwise, the oil phase utilized for 3D printing is silicone oil ($\eta_e \sim 30000$ mPa·s) that contains 10 vol% PDMS-NH₂.”

Revised Table 1 in the Main Text:

Table 1 Different formulations and processing of PEDOT hydrogels and their corresponding electronic and mechanical properties									
Formulation Number	PEDOT:PSS concentration (mg mL ⁻¹)	Additive monomer	Monomer concentration (mg mL ⁻¹)	Ionic liquid concentration (wt%)	UV _{$\lambda=395\text{nm}$} irradiation time (min)	Post-treatment (Y/N)	Modulus (kPa)	Strain at break (%)	Conductivity (S m ⁻¹)
Ink 1	5	PEG1000	385	—	5	N	3875.0	76	1.0×10 ⁻²
Ink 2	5	PEG6000	455	—	5	N	195.2	186	9.2×10 ⁻³
Ink 3	5	PEG20000	313	—	5	N	99.0	287	8.5×10 ⁻²
Ink 4	5	PAAM/PEG20000	263/263	—	5	N	161.5	800	1.9×10 ⁻²
Ink 5	9	PEG1000	100	—	2	N	41.2	136	5.6×10 ⁻²
Ink 5'	9	PEG1000	100	—	2	Y	685.0	127	61.8
Ink 6	9	PEG1000	100	2	2	N	33.2	96	6.8×10 ⁻¹
Ink 6'	9	PEG1000	100	2	2	Y	54.4	130	302.0
Ink 7	9	PEG20000	100	2	2	N	7.3	263	2.0×10 ⁻¹
Ink 7'	9	PEG20000	100	2	2	Y	26.8	275	163.4

All hydrogel samples were printed using a $\Phi 1.6$ mm nozzle with 0.5 m min⁻¹ print head speed and 1.0 mL min⁻¹ ink flow rate.
All hydrogel samples were printed in oil phase (10 vol% PDMS-NH₂ in silicone oil, $\eta_0 \sim 30000$ mPa·s).

The quality of the work and the high standard of the manuscript is at a level suitable for publication in Nature Communications. I recommend publication after the authors has considered the comment regarding the cellular aspect and corrections listed above.

We thank the reviewer again for sharing the thoughtful feedback, and we hope we have addressed these concerns to the satisfaction.

REVIEWERS' COMMENTS

Reviewer #1 (Remarks to the Author):

The reviewer comments are well addressed.

Reviewer #2 (Remarks to the Author):

The authors have revised the manuscript completely. It is recommended for publication in Nature Communications.

Reviewer #3 (Remarks to the Author):

I thank the authors for addressing my concerns in full.

Reviewer #1:

The reviewer comments are well addressed.

Author Response:

We thank the reviewer for their positive evaluation of our revised manuscript.

Reviewer #2:

The authors have revised the manuscript completely. It is recommended for publication in Nature Communications.

Author Response:

We thank the reviewer for their positive evaluation of our revised manuscript.

Reviewer #3:

I thank the authors for addressing my concerns in full.

Author Response:

We thank the reviewer for their positive evaluation of our revised manuscript.